



# Southern Ocean polynyas and dense water formation in a high-resolution, coupled Earth System Model

Hyein Jeong[1,2], Adrian K. Turner[3], Andrew F. Roberts[3], Milena Veneziani[3], Stephen P. Price[3], Xylar S. Asay-Davis[3], Luke P. Van Roekel[3], Wuyin Lin[4], Peter M. Caldwell[5], Jonathan D. Wolfe[3], and Azamat Mametjanov[6]

[1]Institute of Ocean and Atmospheric Sciences (IOAS), Hanyang University, Ansan, South Korea
[2]Department of Ocean Science and Technology, Hanyang University, Ansan, South Korea
[3]Los Alamos National Laboratory, Los Alamos, New Mexico, USA
[4]Brookhaven National Laboratory, Upton, New York, USA
[5]Lawrence Livermore National Laboratory, Livermore, California, USA
[6]Argonne National Laboratory, Lemont, Illinois, USA

**Correspondence:** Hyein Jeong (hijeong820310@gmail.com)

**Abstract.** Antarctic Bottom Water is an important component of Earth's climate system. Its formation occurs through ocean-atmosphere-sea ice flux interactions in coastal and open ocean polynyas around Antarctica. In this paper, we investigate Antarctic dense water formation in the high-resolution version of the Energy Exascale Earth System Model (E3SM-HR). The model is able to reproduce the major Antarctic coastal polynyas, though they are smaller in area compared to observations. E3SM-HR

5    also simulates several occurrences of open-ocean polynyas (OOPs) in the Weddell Sea, at a higher rate than what the last 50 years of satellite sea ice observational record suggests, but similarly to other high-resolution Earth System Model simulations. Furthermore, the densest water masses in the model are formed within the OOPs, rather than on the continental shelf, as is typically observed. Biases related to the lack of dense water formation on the continental shelf are associated with overly strong atmospheric polar easterlies, which lead to a strong Antarctic Slope Front and hence too little communication between on and

10   off continental shelf water masses. Strong polar easterlies also produce excessive southward Ekman transport, causing a build-up of sea ice over the continental shelf and enhanced ice melting in the summer season. This in turn produces water masses on the continental shelf that are overly fresh and less dense relative to observations. Our results indicate that the large-scale polar atmospheric circulation around Antarctica must be accurately simulated in models to properly reproduce Antarctic dense water formation.

## 1 Introduction

Coastal polynyas are areas of ocean surface water or thin, newly formed sea ice surrounded by coastline, ice shelves, or consolidated, thick sea ice (Kusahara et al., 2010). They form by divergent sea ice motion, usually driven by strong offshore





winds, and are often associated with coastal features that block the advection of ice from upstream of the polynyas (Roberts
et al., 2001; Williams et al., 2007; Tamura et al., 2016). Although small in area compared to the total sea ice zone (<3%;
Roberts et al., 2001), these coastal polynyas play an important role in climate by (i) transporting heat from the ocean to the
atmosphere, affecting mesoscale atmospheric motion (Morales Maqueda et al., 2004; Minnett and Key, 2007), and (ii) strongly
modifying water masses through brine rejection resulting from high rates of sea ice production and surface cooling (Williams
et al., 2007). The latter is a key mechanism of cold dense shelf water (DSW) formation, which leads to the formation of
Antarctic Bottom Water (AABW) by downslope transport and mixing with ambient water masses on the continental slope
(Williams et al., 2016). AABW production is a key element of the Earth's climate system (Orsi et al., 1999; Johnson, 2008;
Marshall and Speer, 2012; Williams et al., 2016) as an important sink for carbon dioxide and heat from the atmosphere (Sigman
and Boyle, 2000).

Due to logistical difficulties associated with in-situ observation, sea ice production in coastal polynyas and its interannual
to decadal variability are not well understood or characterized (Ohshima et al., 2016). Therefore, methods to estimate sea ice
production on large scales have been developed using heat flux calculations based on satellite microwave radiometer observa-
tions (Tamura et al., 2006, 2008; Nihashi and Ohshima, 2015; Ohshima et al., 2016). Through the mapping of coastal polynyas
and sea ice production around Antarctica, it has been suggested that a strong link exists between sea ice production and bottom
water formation and its variability (Ohshima et al., 2013, 2016).

The strong link between sea ice production and bottom water formation is the result of DSW flowing across the continental
shelf break downstream of high sea ice production regions (e.g., coastal polynyas). That is, when water masses on the conti-
nental shelf are cold and dense due to dense water formation and shallow convection, a "dense shelf" condition is established
(Thompson et al., 2018). In this situation, DSW can be exported to become bottom water after mixing with ambient water
masses on the continental slope during its descent. The dense shelf condition is associated with a westward-flowing Antarctic
Slope Current (ASC) and a distinctive V-shaped isopycnal frontal structure (Whitworth et al., 1998), enabling onshore trans-
port of Circumpolar Deep Water (CDW) and export of DSW underneath the V-shaped front. The V-shaped frontal structure
is considered to be critical for water mass modification to occur on the continental slope and in setting the correct properties
of AABW that accommodates both an onshore transport of CDW and the export of DSW (Thompson et al., 2018). However,
the westward flowing ASC and distinctive V-shaped front, are not well represented in many fully coupled general circulation
models (GCMs) due to their under-resolving of the steep topography of the Antarctic continental slope and/or model biases
(Dinniman et al., 2016; Thompson et al., 2018).

Numerical models with a good representation of processes in coastal polynyas are also useful tools for improving our
understanding of AABW formation (Kusahara et al., 2010). In previous GCMs with low horizontal resolution, it has been
difficult to explicitly capture coastal polynyas and their impacts on the ocean due to their small areal extent (Stössel and
Markus, 2004; Kusahara et al., 2010). Therefore, studies have also been conducted using high-resolution, regional, coupled
ocean and sea ice models. For example, sea ice processes and DSW formation have been investigated in the Mertz Glacier
polynya (Marsland et al., 2004), Ronne Ice shelf polynya (Årthun et al., 2013), and along the East Antarctic coast (Kusahara



et al., 2010) using coupled ocean and sea ice models. Because sea ice processes in coastal polynyas result from strongly coupled atmosphere, ocean, and sea ice phenomena, fully coupled modeling is needed in addition to high resolution.

Unfortunately, bottom water formation driven by coastal polynyas is notably difficult to represent in GCMs (Aguiar et al., 2017). Instead, GCMs often create AABW through an alternative mechanism, namely open-ocean deep convection (Heuzé et al., 2013; Azaneu et al., 2014; Aguiar et al., 2017). Often when ocean stratification decreases in ocean general circulation models, open-ocean, deep convection occurs in the Southern Ocean, allowing heat transport to the ocean surface from the deeper ocean (Aguiar et al., 2017). This heat transfer creates an ice-free region: an open-ocean polynya (OOP). In the OOP, 60    saline deep waters release sensible heat to the atmosphere, creating AABW by cooling (Killworth, 1983; Aguiar et al., 2017).

Recently, the U.S. Department of Energy (DOE) has developed a variable resolution, fully coupled Earth system model (ESM), the Energy Exascale Earth System Model (E3SM; Golaz et al., 2019; Petersen et al., 2019; Rasch et al., 2019; Lee et al., 2019). The high-resolution, fully coupled version of E3SM (E3SM-HR; Caldwell et al., 2019), which is being run with the purpose of participating in the High Resolution Intercomparison Project (HighResMIP v1.0) for CMIP6 (Haarsma et al., 65    2016), is used in the present study. Unsurprisingly, E3SM-HR reproduces Antarctic coastal polynyas and OOPs with much higher frequency than the standard resolution of E3SM.

In this study, we assess the representation of both types of polynyas in E3SM, compare them with available observational data, and investigate the formation of dense water masses in each type of polynya. In Section 2, we briefly describe E3SM and the reference data sets used for model validation. In section 3, we analyze the representation of Antarctic coastal and 70    open-ocean polynyas compared to available data sets. Section 4 investigates water mass formation in both polynya types, and in sections 5 and 6, we present a discussion, summary, and conclusions from this study.

## 2   Data and methodology

### 2.1   The Energy Exascale Earth System Model

For this study, we use both the high- and low-resolution versions of E3SM v1 (E3SM-HR and E3SM-LR, respectively): these 75    simulations are fully coupled and are run with greenhouse gas conditions in the atmosphere that are fixed at 1950 levels. E3SM v1 consists of atmosphere, land, river, ocean, and sea ice components communicating via a flux coupler (cpl7; Craig et al., 2012). The atmospheric component, the E3SM Atmospheric Model (EAM; Xie et al., 2018; Rasch et al., 2019; Golaz et al., 2019; Caldwell et al., 2019), uses a spectral-element atmospheric dynamical core (Caldwell et al., 2019). The land component is the E3SM Land Model (ELM), which is a slightly revised version of the Community Land Model version 4.5 (CLM4.5; 80    Golaz et al., 2019). The river component is the newly developed Model for Scale Adaptive River Transport (MOSART; Li et al., 2013, 2015; Golaz et al., 2019). The ocean and sea ice components of E3SM v1 are based on the Model for Prediction Across Scale (MPAS) modeling framework (Ringler et al., 2013; Petersen et al., 2019) and share the same unstructured, horizontal grid. The vertical grid of the ocean model is a structured, z-star coordinate (Petersen et al., 2015; Reckinger et al., 2015). While E3SM-HR does not use the Gent-McWilliams (GM; Gent and Mcwilliams, 1990) mesoscale eddy parameterization, E3SM-LR 85    does, with a constant GM bolus eddy diffusivity of $1,800 \text{ m}^2 \text{ s}^{-1}$. A more detailed description of the E3SM v1 ocean and sea





ice components is available in Petersen et al. (2019). To identify the effect of model resolution on simulation results, we use the same tuning parameters on both high and low-resolution E3SM simulations, as described in Caldwell et al. (2019).

Table 1 describes the horizontal and vertical resolution for each model component. The atmosphere and land models share the same horizontal grid, having 25 km and 110 km horizontal spacing for E3SM-HR and E3SM-LR, respectively. The atmospheric model uses a hybrid, sigma-pressure coordinate with 72 vertical layers and a top of the atmosphere at approximately 60 km for both the low- and high-resolution configurations. The land model uses 15 vertical levels for both simulations. For E3SM-HR, the ocean/sea ice mesh features a horizontal resolution varying between 16 km at the equator and 8 km near the poles, and 80 ocean vertical levels with spacing ranging between 2 m at the surface and ∼150 m at depth, following Stewart et al. (2017) to resolve the first and second baroclinic modes in the open ocean. For E3SM-LR, the ocean/sea ice mesh has resolution varying between 30 and 60 km, and 60 ocean vertical levels, with layer thickness varying between 10 m at the surface and 250 m in the deep ocean. The river component employs a latitude and longitude grid with uniform grid spacing in both directions of 0.125° and 0.5° for high- and low-resolution, respectively. For more detailed information on time steps and coupling frequency for each component model, refer to Table 2 in Caldwell et al. (2019).

Before performing fully coupled simulations, we run the ocean model in standalone mode for 1- and 3-months for E3SM-LR and E3SM-HR, starting from rest and the Polar Hydrographic Climatology (PHC; Steele et al., 2001) temperature and salinity profiles, to slightly spin up the velocity field and damp out high-velocity waves. The results of the stand-alone ocean simulation are used as the initial condition for an ocean/sea ice only, three-year simulation using the Common Ocean Reference Experiment version 2 (COREv2; Large and Yeager, 2009) protocol for interannually varying atmospheric forcing. The sea ice is initialized with a uniform thickness of 1 m at all locations south of 70°S and north of 70°N. Finally, the model state from the end of this simulation is used as the initial condition for the ocean and sea ice components in fully coupled simulations. The atmospheric initial condition is obtained from an earlier high-resolution E3SM simulation run with fixed Sea-Surface Temperature (SST). The land initial condition is interpolated from year 1950 of a low-resolution E3SM v1 CMIP simulation (Golaz et al., 2019). With these initial conditions, we run E3SM in a fully coupled model for 50 years at both high- and low-resolution for comparison. We use results from the last 30 years of each simulation for the analysis presented here.

## 2.2 Ocean, sea ice, and atmosphere state estimates

Before investigating the dense water formation in Antarctic coastal and open-ocean polynyas, we evaluate E3SM's representation of both types of polynyas. To do this, we compare E3SM results to available data sets including direct observations, reanalyses, and interpolated climatologies for the atmosphere, ocean, and sea ice in the Southern Ocean. For the evaluation of sea ice, we use sea ice concentration from the NOAA/NSIDC Climate Data Record of Passive Microwave Sea Ice Concentration version 3 (NCDR; Peng et al., 2013) and sea ice production from Nihashi and Ohshima (2015), which is derived from data from the Advanced Microwave Scanning Radiometer for Earth Observing System (AMSR-E). For the evaluation of oceanic properties, we utilize the Southern Ocean State Estimate (SOSE; Mazloff et al., 2010), a state-of-the-art, data-assimilation product that incorporates millions of ocean and sea ice observations while maintaining dynamically consistent ocean state variables. Given the sparsity of observations in many regions around Antarctica, SOSE offers a comprehensive, physically based





estimate of ocean properties that would otherwise be entirely uncharacterized. Specifically, we use SOSE data set with $\frac{1}{6}^\circ$ horizontal resolution, spanning from 2005 to 2010. We also use the World Ocean Atlas 2018 (WOA18; Locarnini et al., 2018; Zweng et al., 2018), which provides a global data product of subsurface ocean temperature and salinity. For the evaluation of the atmospheric state over the Southern Ocean, we use zonal and meridional wind velocities at 10 m from the European Center for Medium-range Weather Forecasts (ECMWF) ERA5 reanalysis product (Hersbach et al., 2019). It should be noted here that

the ocean, sea ice, and atmosphere data sets described above represent present-day conditions, whereas the E3SM simulations are representative of model conditions for the 1950s. We compare monthly averaged products from both present-day observations and E3SM simulations, noting that the horizontal resolution of observational data sets – varying between 16 km and 25 km – differs from E3SM simulation output.

### 2.3 Definition of Antarctic coastal and open-ocean polynyas

We define coastal polynyas in E3SM model output as areas with modeled sea ice that is "thin", as defined by having a thickness less than 20 cm, and surrounded by coastline and/or consolidated thick sea ice during the freezing season (from March to October). This is consistent with the definition used in several observational studies (e.g., Tamura et al., 2006, 2008; Nihashi and Ohshima, 2015; Ohshima et al., 2016). Thin sea ice is a poor thermal barrier and allows one to two orders of magnitude larger heat loss to the atmosphere than thick ice cover (Maykut, 1978; Nihashi and Ohshima, 2015). Therefore, Antarctic coastal

polynyas are areas of very high sea ice production (Markus et al., 1998; Renfrew et al., 2002; Tamura et al., 2008; Nihashi and Ohshima, 2015). In contrast to coastal polynyas, Antarctic open-ocean polynyas occur far from the coast in the middle of the sea ice pack during the winter season. These polynyas are generally observed in conjunction with deep-convection events in the ocean that lead to direct interaction between mid-depth and surface ocean waters. They are also characterized by a strong positive ocean-to-atmosphere heat flux (Kurtakoti et al., 2018). These processes result in areas of ice-free ocean. We define

OOPs in E3SM's model output in the same way as previous studies (e.g., Kurtakoti et al., 2018), i.e. as areas within ice pack where modeled open ocean sea ice concentration is less than 15% during the freezing season.

## 3 Antarctic coastal and open-ocean polynyas

### 3.1 Coastal polynyas

In this section, we investigate the fidelity of Antarctic coastal polynyas simulated in E3SM-HR (and E3SM-LR) by comparing

model output with available observational data. Cold katabatic winds blowing off Antarctica create sea ice in the Southern Ocean during the freezing season (e.g., as shown for E3SM-HR in Fig. 1a and b). Evident in this figure is that there is more sea ice production near the Antarctic coast compared to further offshore. Cold, salty waters generated by high sea ice production near the Antarctic coast are likely to contribute to AABW formation through transport down the continental slope (Fig. 1b). Nihashi and Ohshima (2015) defined 13 major Antarctic coastal polynyas and investigated the polynya area and accumulated

sea ice volume of these using AMSR-E (See Table 3). In Fig. 1c, we compare E3SM-HR's accumulated sea ice volume as a



function of longitude from March to October from Antarctic coastal polynyas with a satellite estimate (AMSR-E; Nihashi and Ohshima, 2015). The AMSR-E data shows that the highest rates of sea ice production occur along the East Antarctic coast, while the Ross Sea at around 180° longitude (Ross Ice Shelf Polynya (RISP)) has the largest total sea ice production. The analysis of Ohshima et al. (2013) suggests that the Cape Darnley Polynya (CDP), located west of the Amery Ice Shelf (around

70°E), has the second-highest ice production around the Antarctic after the RISP. With some exceptions that we will explore below, E3SM-HR generally does well at representing the accumulated sea ice volume in coastal Antarctic polynyas, especially in high sea ice production areas along the East Antarctic coast. Since sea ice production in coastal polynyas is largely caused by the latent heat flux released to the atmosphere, we also find that E3SM-HR shows relatively higher latent heat flux released from the ocean to the atmosphere in East Antarctica compared to the West Antarctic (Fig. 1a). We further find that, over the

course of the simulation, E3SM-HR displays interannual variability in total sea ice production from Antarctic coastal polynyas of up to 43% (Fig. 1d). This suggests that E3SM-HR may be useful for the analysis of Antarctic coastal polynya variability.

In Fig. 2 we plot the mean coastal polynya area and accumulated sea ice volume during the March to October freezing season for each of the 13 polynyas indicated in Fig. 1b, which are simulated by both the high and low-resolution versions of E3SM, and compare them with the observed estimates. In general, E3SM-LR shows very little polynya area or sea ice volume

production over the defined coastal polynya region. This is a common limitation in ESMs with low horizontal resolution due to coastal polynyas' small areal extent (Stössel and Markus, 2004; Kusahara et al., 2010). The E3SM-HR shows significant improvement in the representation of coastal polynya area and sea ice volume production compared to E3SM-LR. Yet, its sea ice volume production in several coastal polynyas such as the Dalton, Dibble, and Amundsen Polynyas (DaP, DiP, AP, respectively) is much smaller than observed. In reality, these coastal polynyas are strongly affected by the presence of landfast

ice, a feature not yet represented in E3SM. Landfast ice is stationary sea ice attached to coastal features such as the shoreline and grounded icebergs (Nihashi and Ohshima, 2015). Several previous studies suggested that landfast ice and glacier tongues play an important role in the formation of some coastal polynyas by blocking sea ice advection of upstream sea ice into the polynya and thereby facilitating divergent motion (Nihashi and Ohshima, 2015; Bromwich and Kurtz, 1984). Therefore, a representation of landfast ice may be needed to more accurately capture coastal polynya characteristics. When we compare the

summation of polynya area and sea ice volume production for polynyas not strongly associated with landfast ice (Fig. 2o), it can be seen that E3SM-HR does reasonably well at reproducing sea ice volume production compared to AMSR-E observations, despite generally underestimating polynya area.

## 3.2   Open-ocean polynyas

The first observed OOP, the Maud Rise Polynya (MRP), was detected during the period June to October 1973 and reappeared

during the period from June to August 1974. The MRP is relatively small in area and occurs above the Maud Rise seamount in the eastern Weddell Sea (Kurtakoti et al., 2018). In the winters of 1974 and 1975, the MRP extended westward into the central Weddell Sea, initiating a Weddell Sea Polynya (WSP; Kurtakoti et al., 2018). The WSP is a relatively large, open-ocean polynya in the central Weddell Sea, having a maximum ice-free area of ∼250,000 km². More recently, the MRP has been observed intermittently in August 2016 and more consistently during the winter of 2017 (Fig. 3a–c). The 2017 MRP appeared





in September and had its maximum extent in November of that year. While similar results were reported by Dufour et al. (2017), E3SM-LR does not exhibit OOPs at any point in the simulation, E3SM-HR does produce MRPs, an example of which is shown in Fig. 3d–f for model year 54, where relatively low sea ice concentration can be seen around 0° longitude. As sea ice concentration decreases in the polynya area, the ocean and atmosphere can interact directly. This causes the release of sensible heat from the ocean to the atmosphere (Fig. 3g–i), thus allowing for dense-water formation in E3SM-HR.

Unlike in the real world, OOPs occur frequently in some fully coupled ESM simulations with high-resolution (e.g., Kurtakoti et al., 2018). Even in high-resolution ocean reanalysis products, there are several cases of OOPs (for instance, in 2005 in SOSE and in 2004, 2007, and 2010 in ECCO2) that do not appear in satellite observations (Aguiar et al., 2017). The convective activity that accompanies OOPs is associated with decreased stability within the water column, which can be caused by buoyancy changes in surface or deep waters (Azaneu et al., 2014). In addition, the strong polar cyclonic gyre that exists in the Weddell

Sea may be a factor in preconditioning this convection process, since it leads to shoaling of the pycnocline and circulation that strongly interacts with the bathymetry near Maud Rise (Gordon and Huber, 1990; Azaneu et al., 2014). Perhaps because of an overly strong cyclonic gyre related to a sub-polar low that is too deep (see Section 4.2a), E3SM-HR produces OOPs in 18 of the 30 total simulation years (see Fig. 4), far more frequently than observed. We define the simulated OOP years as those in which either or both MRP and WSP occur, or embayment-like features in sea ice concentration occur in the Weddell Sea (Fig. 4c–f).

During non-OOP years, most of the Weddell Sea is covered with sea ice from September to November (Fig. 4a). During OOP years, however, localized ocean convection associated with enhanced upper ocean salinity and a strong Taylor column near the Maud Rise seamount produces ventilation of subsurface warmer waters, which in turn inhibits sea ice formation within the area where deep convection occurs. Once an OOP occurs in a model simulation, it tends to occur again in subsequent years (Fig. 4g) because of the ventilation of salty/high-density waters that occurs within the deep convection area (Kurtakoti

et al., 2020). In the following sections, in order to investigate the dense water formation associated with OOP events, which potentially contribute to AABW formation, we compare simulation years with OOP events to none OOP years lacking OOPs.

## 4 Dense water formation

### 4.1 Water mass transformation in OOPs and continental shelves

In the previous section, we found that E3SM-HR can produce not only open-ocean but also coastal polynyas. This implies

that E3SM-HR may allow for dense water-mass formation in the Southern Ocean via both types of polynya. Several modeling studies, however, have suggested that ESMs often create bottom water mostly through open-ocean deep convection in the Weddell Sea rather than the more realistic mechanism of dense water formation by sea ice production in coastal polynyas (Heuzé et al., 2013; Azaneu et al., 2014; Aguiar et al., 2017). In this section, we apply a water-mass transformation (WMT) analysis to the last 30 years of E3SM-HR simulation in the Weddell Sea region over both the continental shelf and the open

ocean, to find out which polynya type predominantly produces dense water masses in E3SM-HR. The WMT analysis, first introduced by Walin (1982), quantifies the relationship between the thermodynamic transformation of water mass properties within an ocean basin and the net transport of those same properties into or out of the basin. This relationship has been used



to characterize the thermodynamic processes that sustain the Southern Ocean overturning in models (Abernathey et al., 2016). A more detailed description of WMT analysis can be found in Groeskamp et al. (2019). Jeong et al. (2020) discuss previous applications of WMT to the analysis of low-resolution E3SM simulations.

In Fig. 5a, we show the mean WMT rate over the Southern Ocean produced in E3SM-HR by all surface fluxes combined (positive WMT rates indicate that water masses become denser, implying buoyancy loss; negative WMT rates indicate that water masses become lighter, implying buoyancy gain). Because we only focus on the WMT rate during the freezing season, from March to October, positive WMT rates dominate over most of the Southern Ocean while slightly negative transformation rates are found in some regions. Over the continental shelf, the region delimited by the 1000 m isobath in Fig. 5a, we find strong positive WMT rates, which are the direct result of high sea ice production there (Fig. 1b). The mean WMT rates in latitude and longitude space (Fig. 5a) can be converted into neutral density space (Fig. 5b). The maximum WMT rates over the whole Southern Ocean ($\approx 20$ Sv) is found at a neutral density of $27.6 \, \mathrm{kg \, m^{-3}}$. In the very high density ranges above $28.0 \, \mathrm{kg \, m^{-3}}$, another region of positive transformation rate can be seen with a maximum WMT rate of $3.0 \, \mathrm{Sv}$ at a neutral density of $28.2 \, \mathrm{kg \, m^{-3}}$. Positive transformation rates above $28.0 \, \mathrm{kg \, m^{-3}}$ are consistent with AABW formation; Orsi et al. (1999) define Southern Ocean bottom water as having neutral densities higher than $28.27 \, \mathrm{kg \, m^{-3}}$. As this high-density transformation rate in E3SM-HR is entirely due to WMT over the Weddell Sea, rather than on the continental shelf (compare green and grey curves in Fig. 5b), the simulated bottom water is produced through open-ocean convection in the Weddell Sea only, similarly to previous studies (Heuzé et al., 2013; Azaneu et al., 2014; Aguiar et al., 2017). Over the continental shelf, we also see positive transformation rates, but these occur at relatively light densities, with an average neutral density of $27.5 \, \mathrm{kg \, m^{-3}}$, whereas no transformation occurs at densities higher than $28.0 \, \mathrm{kg \, m^{-3}}$.

When we consider the WMT rate contributed by several types of surface fluxes separately over the Antarctic continental shelf and over the Weddell Sea (Fig. 5c and d), the positive transformation rate over the continental shelf results almost entirely from brine rejection by sea ice production. The positive transformation rate over the deep Weddell Sea, meanwhile, is mostly due to the surface heat release from the ocean to the atmosphere, especially at relatively high neutral densities above $28.0 \, \mathrm{kg \, m^{-3}}$, which is consistent with the strong ocean-to-atmosphere exchanges that occur in OOPs. We confirm, therefore, that the coastal and open-ocean polynyas simulated in the E3SM-HR have different WMT mechanisms. However, and as described above, the simulation produces water masses at relatively low-density levels on the continental shelf, despite the fact that the model does reasonably well at reproducing a large amount of sea ice formation in coastal polynyas during the freezing season. To investigate this apparent inconsistency, in the next section, we examine E3SM-HR's hydrographic characteristics in the Southern Ocean.

## 4.2 Hydrographic characteristics of continental shelves

As the polar ocean is highly controlled by atmospheric dynamics and/or thermodynamics, while freshwater dynamics also make an important contribution (Hattermann, 2018; Moorman et al., 2020), we start by comparing E3SM-HR's atmospheric surface (10 m) winds with the ERA5 reanalysis data (Fig. 6a,b). We find that, compared to ERA5, E3SM-HR has stronger atmospheric surface winds not only along with the Antarctic continental shelf but also in the interior Southern Ocean, which is



consistent with an overly deep (simulated) sub-polar low-pressure system at around 60–70°S (figure not shown). These overly strong atmospheric winds may produce a strong Antarctic Slope Current (ASC). The ASC is a coherent circulation feature that rings the Antarctic continental shelf and regulates the flow of water toward the Antarctic coastline (e.g., Jacobs, 1991; Thompson et al., 2018). Indeed, as seen in Fig. 6c,d, we find a very strong ASC in the E3SM-HR simulation, with a speed more than double that in SOSE ($0.16\,\mathrm{m\,s^{-1}}$ in SOSE compared with $0.34\,\mathrm{m\,s^{-1}}$ in E3SM-HR at 0°). As we discuss in more detail below, this is likely a result of the strong easterly winds over the continental shelf.

Thompson et al. (2018) defined three Antarctic continental shelf types – fresh, dense, and warm shelves – based on the type of ocean stratification present on the shelf/slope and specific oceanic processes occurring on the shelf. A fresh shelf (East Antarctic sector) is characterized by low salinity on the shelf and strong easterly winds at the coast and offshore, which produce onshore Ekman transport and downwelling. These processes establish downward sloping isopycnals against the continental slope and a surface intensified front, that manifests itself as a strong westward flowing ASC. The stratification associated with fresh shelves is such that warm and saline CDW is prevented from flowing onto the shelf (Thompson et al., 2018). A dense shelf (e.g., the western Weddell Sea and the eastern Ross Sea) occurs when the shelf is relatively cold and dense due to brine rejection as a result of high sea ice formation rates (coastal polynya mechanism). The density distribution is characterized by a "V-shape" near the continental slope, with convection occurring on the onshore side and the Antarctic Slope Front (ASF) still supporting a westward flowing ASC on the offshore side (although weaker than in the fresh-shelf case). A warm-shelf (West Antarctic sector) is characterized by slightly upward sloping isopycnals, which enable easier onshore transport of warm, intermediate depth CDW.

To investigate how these shelf types are reproduced in E3SM-HR, we select three vertical meridional cross-sections; at 12°W (eastern Weddell Sea), 35°W (western Weddell Sea), and 72°W (west coast of Antarctic peninsula), which are representative of the fresh-, dense-, and warm-shelf types, respectively (the locations for each shelf type are indicated in Fig. 6d). We present the results in Figs. 7- 9, comparing the model temperature and salinity profiles with SOSE and with the observational results of Whitworth et al. (1998).

For the fresh-shelf type (Fig. 7), E3SM-HR simulates an ASC that is almost twice as strong as that simulated by SOSE (Fig. 7c,e), with isohaline surfaces strongly tilted downwards in the onshore direction, more so than in the observational results described in Whitworth et al. (1998, compare their Figure 2 with Fig. 7e; note that here isohalines are used as a proxy for isopycnals since salinity is the primary influence on the density distribution at surface pressure and temperature close to the freezing point). As shown in the zonal and meridional wind profiles from the model and ERA5 (Fig. 7a,b), the stronger ASC is associated with easterly winds that are $\approx 2\,\mathrm{m\,s^{-1}}$ stronger than in ERA5.

The dense shelf type occurs downstream of active sea ice formation regions such as the Ross Sea, the western Weddell Sea, the Adélie coast, and Cape Darnley. Cold and salty water masses, induced by brine rejection, flow across the continental shelf break producing AABW. Similar to the fresh shelf type, warm CDW in these regions tilts down toward the seafloor over most of the continental slope. However, as the CDW approaches the shelf break in the dense shelf type, it shoals again (see the "V-shaped" isopycnal surface in Fig. 3b from Thompson et al. 2018). The V-shaped isopycnal surface is critical to water mass transformation and to setting the properties of AABW (Thompson et al., 2018) because it accommodates both an onshore





transport of CDW and the export of DSW. While we do not see the V-shaped isohaline surface in SOSE (Fig. 8c), E3SM-HR also exhibits no V-shaped isohalines, but rather steep isohaline surfaces tilted downward toward the shelf break from offshore, quite differently from observational results at the same location (compare Fig. 8e with Fig. 3 in Whitworth et al. 1998). The

290 E3SM-HR stratification for this case is more similar to the fresh shelf type, with a similar ASC as well (compare Fig. 8d,e with Fig. 7d,e). This is also likely related to stronger model winds compared to ERA5 (Fig. 7a,b), and specifically to a largely easterly component in the model winds that is much reduced in ERA5 north of 74°S. Also, it could indirectly be a result of excessively strong winds advecting freshwater into the Weddell Sea from upstream (Graham and De Boer, 2013), or from the ASF being too strong around the continent and retaining freshwater on the continental shelf (Moorman et al., 2020).

Lastly, the warm shelf type is mainly observed along the coasts of the West Antarctic, particularly in the Amundsen and Bellingshausen Seas. Unlike the fresh shelf type, the ASC is weak or absent over this type of shelf (Thompson et al., 2018). Coastal atmospheric wind speed is also observed to be relatively weaker than in the East Antarctic. As can be seen in Fig. 9c, SOSE has flat isohaline surfaces tilted toward Antarctica, resulting in warm CDW reaching the continental shelf, consistent with observations (see Fig. 10 in Whitworth et al. 1998 or Fig. 3 in Thompson et al. 2018). Again, E3SM-HR simulates

hydrographic conditions that are more similar to the fresh-shelf type, with downward tilted isohalines and a weak westward flowing ASC at the shelf break (Fig. 9e,d). This bias is similarly related to model easterly winds that are twice as strong as those in ERA5 (Fig. 9a).

In summary, as a result of ubiquitous, strong polar easterly winds, we find that E3SM-HR consistently displays only the fresh-shelf type. These overly strong winds also cause overly strong onshore Ekman transport, which in turn affects the advection

of sea ice formed during the winter season. As shown in Fig. 10, E3SM-HR's simulated sea ice extent in the Southern Ocean is much less than that from SOSE in all seasons except for JAS (July-August-September) but sea ice volume is similar or even slightly larger than SOSE's ice volume. This is likely the result of sea ice build up on the continental shelf in E3SM-HR, where sea ice thickness often exceeds 2 m; enhanced poleward surface Ekman transport prevents sea ice from moving to lower latitudes, where it would potentially melt.

Modeled and observed SST and sea-surface salinity (SSS) provide additional information regarding the impacts of these same biases. In Fig. 11 we compare SST and SSS from E3SM-HR with that from SOSE and WOA18. E3SM-HR has a good representation of seasonal SST and SSS variation, but with warmer/fresher and colder/fresher water properties in the summer and winter seasons, respectively, compared to SOSE and WOA18. In summer, the warm, fresh surface water has a neutral density of $26.0\,\mathrm{kg\,m^{-3}}$ in E3SM-HR, compared with more than $27.2\,\mathrm{kg\,m^{-3}}$ in WOA18 and $26.8\,\mathrm{kg\,m^{-3}}$ in SOSE. The

possible external sources of the relatively low summer density in E3SM-HR can be excessive precipitation (E-P), runoff, or sea ice melting. Consistent with the sea ice thickness bias discussed above (Fig. 10), we find that the freshwater flux into the ocean from melting sea ice in E3SM-HR is larger than that of SOSE, especially over the continental shelf and shelf break (Fig. 11b), while precipitation and runoff differences are not significant compared to SOSE. Because of extensive and excessive freshwater inputs from sea ice in E3SM-HR, the annual mean sea-surface neutral density is lighter than WOA18 over the continental shelf

(Fig. 11d and e). We thus conclude that the overly strong easterly winds are also (indirectly) responsible for the relatively lighter water masses forming on the continental shelf in E3SM-HR.





## 5   Discussion

Coupled GCMs are sophisticated tools designed to simulate, understand, and predict the behavior of the Earth's climate system
(Laloyaux et al., 2016). The model validation or assessment of climate phenomena from Coupled GCMs has, however, tradi-
tionally been compared with uncoupled reanalysis data as we have done in this study. We may have over-interpreted biases in
E3SM relative to uncoupled reanalysis products, given that they also have their own biases, in part because they lack physics-
based constraints. Hence, there is a need for methods that are designed to ingest observations of the atmosphere and ocean in a
coupled model in a consistent manner (Laloyaux et al., 2016). Improvements in model capabilities and increases in computing
power are enabling the analysis of Earth's climate system using such fully coupled data assimilation schemes (Penny et al.,
2019) and future assessments of E3SM would greatly benefit from such coupled model reanalysis products.

Whereas the E3SM simulations presented in this paper use the HighResMIP 1950 atmospheric forcing and therefore reflect
the internal variability of a stable climate, the observations with which they are compared are for a transient climate. Moreover,
the observations presented in this study are from 1979 onwards, many from this century, which is at a significantly different
point in the industrial epoch than is represented in our 1950s trace atmospheric model constituents. However, previous studies
(e.g., Santer et al., 2008; Notz et al., 2013; Swart et al., 2015) indicated that the trends or mean values in observations and indi-
vidual model simulations may differ significantly because of random differences in internal variability, rather than differences
in response to the external forcing. In addition, Menary et al. (2018) and Jeong et al. (2020) found that the differences between
preindustrial and present-day control simulations are much smaller than the differences between different model configura-
tions under the same preindustrial or present-day forcing. Finally, we note that E3SM-HR uses an Antarctic land-sea boundary
based on data from the 1990s and 2000s, which includes floating ice tongues that are important for determining the locations
of coastal polynyas (and which are also present in the observations used in this study). For all of these reasons, we feel justified
in using present-day observations as a metric by which to judge our model output.

There are several phenomena known to be important in the physics associated with coastal polynyas that are challenging
to model. Perhaps most importantly, the ASC plays a critical role in Earth's climate system as it affects the large-scale ocean
circulation, the stability of Antarctic ice sheets, and the global carbon cycle (Thompson et al., 2018). The representation of
the ASC in ocean models has been challenging (Dinniman et al., 2016), and it is not well represented in low-resolution GCMs
due to inadequate resolution of the steep topography of the Antarctic continental-shelf slope (Thompson et al., 2018). While
the ASC is largely absent in low-resolution E3SM simulations (Jeong et al., 2020) it has a much improved representation in
E3SM-HR. This is probably due to the higher resolution of simulated surface winds, the use of a higher resolution bathymetry,
and better reproduction of mesoscale eddy activity over the Antarctic continental shelf and slope. The ASC, however, is not
only determined by atmospheric winds and possibly eddy activity but also depends on tidal flow across the continental shelf
break (Stewart et al., 2019), another process currently missing in our ocean model. Similarly, an explicit physical model of
landfast sea ice (e.g., Lemieux et al., 2018) is also absent from the model. The E3SM sea ice dynamics modeled along the coast
also possess no tensile strength, nor interaction with icebergs, both known to be important for shaping coastal polynyas (Fraser





et al., 2012). We intend to explore the impacts of tides, landfast ice, and icebergs on polynya formation in future studies using E3SM.

## 6   Summary and conclusions

In this paper, we have investigated dense water formation in Antarctic coastal and open-ocean polynyas using the fully coupled E3SM-HR simulation. In terms of sea ice area and sea ice volume, we find that E3SM-HR reproduces the main features

of the 13 most important Antarctic coastal polynyas. There are limitations, however, with the representation of several coastal polynyas, which are possibly related to E3SM's current inability to represent landfast ice. This suggests that adding an accurate representation of landfast ice in E3SM-HR will be essential for correctly simulating coastal polynya and for future studies of polynya variability. E3SM-HR also reproduces MRPs, which were observed in the winters of 2016 and 2017 and, prior to that in the 1970s. The model shows good agreement with satellite observations of the 2017 areal extent and time evolution of the

MRP. E3SM-HR frequently produces several other types of OOP events, such as the WSP, a concurrent MRP and WSP, and embayment-shaped polynyas. Aside from the large embayments, all of these were observed in the 1970s.

We investigated dense water formation on the continental shelf and in the Weddell Sea using a WMT analysis. We found significant positive WMT rates on the Antarctic continental shelf, which are entirely due to brine rejection by sea ice formation there. However, the positive WMT rates occur at lower densities than expected. Instead, the densest water masses are formed

in the interior Weddell Sea when heat is transferred from the deep ocean to the surface and then to the atmosphere.

We found that E3SM-HR has overly strong polar easterlies, resulting in a strong ASC separating cold and fresh water nearly everywhere on the continental shelf from warm and salty CDW further offshore. Anomalously strong polar easterlies also lead to overly strong southward surface Ekman transport and sea ice melting on the continental shelf in the summer season. Possibly because of this, the density of seawater on the continental shelf tends to be too low throughout the year in E3SM-HR,

producing a large density contrast between the continental shelf and waters further offshore. Indeed, and as mentioned above, there is no Antarctic Bottom Water formation in the simulated coastal polynyas, despite the fact that the simulated polynyas produce large amounts of sea ice associated with brine rejection. One hypothesis that we hope to investigate in future work is that this occurs because of the low mean densities present on the continental shelf. We suspect that the low mean density is due to melting of excessively thick sea ice in the Summer. The reason for this is that there are overly strong easterlies along

the coast and the associated excessive accumulation of ice along the coast due to Ekman dynamics. On the other hand, in the interior Weddell Sea, E3SM-HR allows for an alternative method of deep ocean ventilation via deep ocean convection, strong air-sea interactions, and dense water formation, consistent with findings from previous modeling studies (Azaneu et al., 2014; Aguiar et al., 2017; Kurtakoti et al., 2018).

The strong polar easterlies in E3SM-HR are due to an overly deep subpolar low-pressure system around 60–70°S, which

also causes an intensification of the cyclonic ocean circulation in the Weddell Sea (Weddell Gyre). Studies of WSPs (e.g., Cheon et al., 2014) suggest that the intensification of the Weddell Gyre is one of the possible pre-conditioning mechanisms for
WSPs, due to the Ekman pumping and raising of isopycnals in the center of the gyre. Thus, overly deep sub-polar low-pressure systems may impact both open-ocean polynya phenomena and the water mass formation occurring in coastal polynyas.

Regardless of the biases described above, we have shown here that fully coupled, high-resolution E3SM simulations can partially reproduce Antarctic coastal polynyas, an important advancement given their importance as a source of AABW formation and that AABW formation processes are currently under- or misrepresented in most ESMs. Silvano et al. (2018) suggest that increased submarine melting of Antarctic ice shelves may reduce AABW formation by offsetting salt fluxes during sea ice formation in coastal polynyas, a change that could then prevent full-depth convection and the formation of dense shelf water. This hypothesis is supported by the simulations and WMT analyses of Jeong et al. (2020), where the impacts of explicitly including

submarine ice shelf melt fluxes in a low-resolution ESM were explored. Future work should focus on better understanding the impacts of both high spatial resolution and sub-ice shelf melting on simulations of Antarctic coastal polynyas, dense water formation, ocean stratification, and the potential feedback between these processes.

*Author contributions.* HJ, AT, AR, and MV designed this study. PC, JW, and AM ran the simulations. HJ made the plots, performed the analysis, and wrote a draft of the manuscript. SP, XA, LV, and WL provided important guidance, while all the authors discussed and revised

the manuscript.

*Competing interests.* The authors declare that they have no conflict of interest.

*Acknowledgements.* This research was supported by the Energy Exascale Earth System Model (E3SM) project, funded by the U.S. Department of Energy Office of Science, Biological and Environmental Research program and the research fund of Hanyang University (HY-2020). E3SM simulations used computing resources from the Argonne Leadership Computing Facility (U.S. DOE contract DE-AC02-06CH11357),

the National Energy Research Scientific Computing Center (U.S. DOE contract DE-AC02-05CH11231), and the Oak Ridge Leadership Computing Facility at the Oak Ridge National Laboratory (U.S. DOE contract DE-AC05-00OR22725), awarded under an ASCR Leadership Computing Challenge (ALCC) award. Peter Caldwell's work at Lawrence Livermore National Laboratory was supported under DOE Contract DE-AC52-07NA27344.



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





**Table 1.** Horizontal and vertical resolution of E3SM-HR and E3SM-LR. In the case of sea ice, vertical levels refer to the number of thickness categories.

|  | Component | E3SM-HR | E3SM-LR |
|---|---|---|---|
| Horizontal spacing | Atmosphere / Land | 25 km | 110 km |
|  | Ocean / sea ice | 8–16 km | 30–60 km |
|  | River | 0.125° | 0.5° |
|  | Atmosphere | 72 | 72 |
|  | Land | 15 | 15 |
| Vertical levels | Ocean | 80 | 60 |
|  | Sea ice | 5 | 5 |
|  | River | 1 | 1 |

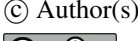



**Table 2.** Atmosphere, ocean, and sea ice state estimation datasets

| Data sets | Variables | Units | Spatial and temporal resolution | Periods | References |
|---|---|---|---|---|---|
| AMSR-E | Sea ice production | $\mathrm{m\,yr^{-1}}$ | 20 km, Monthly | 2003-10 | Nihashi and Ohshima (2015) |
| NCDR | Sea ice concentration | N/A | 25 km, Monthly | 1989-2018 | Peng et al. (2013) |
| SOSE | Sea ice concentration | N/A | 16 km, Monthly | 2005-10 | Mazloff et al. (2010) |
| | Sea ice thickness | m | | | |
| | Temperature | °C | | | |
| | Salinity | psu | | | |
| | Potential denstiy | $\mathrm{kg\,m^{-3}}$ | | | |
| | Zonal velocity | $\mathrm{m\,s^{-1}}$ | | | |
| | Meridional velocity | $\mathrm{m\,s^{-1}}$ | | | |
| WOA18 | Temperature | °C | 25 km, Monthly | 1995-2018 | Locarnini et al. (2018) |
| | Salinity | psu | | | Zweng et al. (2018) |
| ERA5 | Zonal wind at 10 m | $\mathrm{m\,s^{-1}}$ | 25 km, Monthly | 1979-2008 | Hersbach et al. (2019) |
| | Meridional wind at 10 m | $\mathrm{m\,s^{-1}}$ | | | |

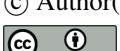



**Table 3.** 13 major Antarctic coastal polynyas in this study

| Acronym | Polynya name | Polynya area ($10^3 \, \mathrm{km}^2$) | Sea ice production ($10^{10} \, \mathrm{m}^3$) |
|---------|--------------|------------------|-----------------------|
| CDP | Cape Darnley | 10.3±3.7 | 13.4±1.3 |
| MBP | Mackenzie Bay | 3.9±2.1 | 6.0±0.6 |
| BaP | Barrier | 6.0±2.7 | 6.2±0.7 |
| SP | Shackleton | 7.5±3.6 | 8.4±0.8 |
| VBP | Vincennes Bay | 6.3±2.2 | 6.4±0.5 |
| DaP | Dalton | 3.7±2.0 | 3.5±0.4 |
| DiP | Dibble | 5.5±2.3 | 5.7±0.9 |
| MP | Mertz | 9.7±4.4 | 13.2±1.9 |
| TNBP | Terra Nova Bay | 3.6±2.1 | 5.9±0.6 |
| RISP | Ross Ice Shelf | 17.7±10.6 | 30.0±2.2 |
| AP | Amundsen | 7.7±3.6 | 9.0±1.4 |
| BeP | Bellingshausen | 4.9±2.8 | 5.5±1.2 |
| RONP | Ronne Ice Shelf | 2.3±2.7 | 3.8±1.6 |



**Figure 1.** (a) Mean latent heat flux from the sea surface to the atmosphere (shading) and surface wind vectors during the freezing season (March–October) in E3SM-HR. Units of reference for the wind vector are $\mathrm{m\,s^{-1}}$. (b) Mean sea ice production rate during the freezing season (shading) and 1000 m isobath (contour) in E3SM-HR. Locations of individual coastal polynyas are shown. Definitions of abbreviations can be found in Table 3. (c) Accumulated sea ice volume as a function of longitude from Antarctic coastal polynyas from March to October from E3SM-HR (magenta) and AMSR-E (gray). (d) Total sea ice volume production by year for all Antarctic coastal polynyas for model years 26 to 55.



**Figure 2.** (a–m) AV (Area-Volume) diagrams of average polynya area (x-axis) and mean annual volume of sea ice production (y-axis) during the freezing season from March to October for the 13 major Antarctic coastal polynyas. The black bar represents the standard deviation in the annual mean area for each coastal polynya and the grey bar represents the standard deviation of sea ice production for each coastal polynya. (n) Summation of polynya area and sea ice volume production for the 13 major coastal polynyas. (o) Summation of polynya area and sea ice volume production for the 9 major coastal polynyas where landfast ice does not play a significant role (BaP, DaP, DiP, and AP are excluded).



**Figure 3.** Observed sea ice concentration during the month of (a) September, (b) October, and (c) November from NCDR (the observation year 2017) over the Southern Ocean. Simulated sea ice concentration during the month of (d) September, (e) October, and (f) November from E3SM-HR (model year 54) over the Southern Ocean. Simulated sensible heat flux from E3SM-HR during the month of (g) September, (h) October, and (i) November from model year 54. Note that the area where sea ice concentration is larger than 15% is masked out.



**Figure 4.** Composite mean of simulated sea ice concentration in E3SM-HR from September to November for years with (a) no OOP (normal year), (b) OOPs (MRPs+WSPs+Embayments), (c) Maud Rise Polynyas, (d) Weddell Sea Polynyas, (e) Maud Rise and Weddell Sea Polynyas, and (f) Embayment shape cases. The numbers in parentheses are the number of years used to compute each composite analysis (years used for the composites can be found in (g)). The OOP years consist of all the cases of (c) Maud Rise polynyas, (d) Weddell Sea polynyas, (e) Maud Rise and Weddell Sea polynyas, and (f) Embayment shapes. The pink boxes in (a) and (b) indicate the area used to calculate the WMT rate in Section 4. (g) Averaged sea ice extent by E3SM model year from September to November over the Weddell Sea (pink box in (a)). The normal year is denoted by a gray color and the OOP type is denoted by using 4 different colors.





**Figure 5.** (a) Thirty-year mean water-mass transformation (WMT) rate from March to October (freezing season) by total surface fluxes, including surface heat and freshwater fluxes. The green bold contour marks the 1000 m isobath (which defines the continental shelf here) and the pink box identifies the Weddell Sea as defined in this study. (b) WMT rates with respect to neutral density class summed over the whole Southern Ocean (gray), continental shelf (green), and Weddell Sea (pink). Note that the green and pink curves do not sum to the gray curve. (c) and (d) show WMT rates driven by the total surface flux and its components integrated over the continental shelf and over the open Weddell Sea, respectively. WMT rate driven by the total surface flux (solid black) consists of those induced by surface heat flux (solid red), freshwater flux from sea ice formation (solid blue), sea ice melting (dashed orange), and E-P-R (dashed green; Evaporation, Precipitation and Runoff).



**Figure 6.** Annual-mean atmospheric surface winds (at 10m) and streamlines from (a) ERA5 and (b) E3SM-HR. Annual mean sea-surface ocean currents from (c) SOSE and (d) E3SM-HR. Shading represents speed of atmospheric winds and ocean currents. Vector units are $\mathrm{m\,s^{-1}}$. The locations for each shelf type are indicated by "W", "D", and "F" in (d) for representing warm, dense, and fresh shelves, respectively.



**Figure 7.** (a, b) Atmospheric zonal and meridional wind speed at 10 m and 12°W from ERA5 and E3SM-HR with respect to latitude. Vertical cross-section at 12°W of zonal ocean velocity from (c) SOSE, and (e) E3SM-HR. Negative zonal velocity represents westward flow while positive zonal velocity represents eastward flow. Vertical cross section at 12°W of ocean temperature (shading) and salinity (contours) from (d) SOSE and (f) E3SM-HR. Salinity contours start at 34.0 psu and are separated by 0.1 psu intervals. All figures here are annual averages. 12°W typically represents the fresh shelf type in observations.





**Figure 8.** As in Fig. 7 but for 35°W. This longitude typically represents the dense shelf type in observations.





**Figure 9.** As in Fig. 7 but for 72°W. This longitude typically represents the warm shelf type in observations.



**Figure 10.** Seasonal mean sea ice thickness during JFM (January-February-March), AMJ (April-May-June), JAS (July-August-September), and OND (October-November-December) from (a,c,e,g) SOSE and (b,d,f,h) E3SM-HR. The region where sea ice concentration is less than 15% is masked out. The yellow contour line denotes the 1000 m isobath. Sea ice extents and volumes for each season are also displayed.





**Figure 11.** (a) Seasonal and area-averaged Sea-Surface Temperature (SST) and Sea-Surface Salinity (SSS) from E3SM-HR, SOSE, and WOA18 over the continental shelves for the season of January-February-March (JFM, Summer), April-May-June (AMJ, Autumn), July-August-September (JAS, Winter), and October-November-December (OND, Spring). Dashed lines on the T-S diagram are neutral density contours. Seasonal mean freshwater flux into the ocean by sea ice melting for JFM from (b) SOSE and (c) E3SM-HR. Annual mean sea surface neutral density from (d) WOA18 and (e) E3SM-HR. Red (b,c) and gray (d,e) bold contours represent the 1000 m isobath.