# Peer review of "Southern Ocean polynyas and dense water formation in a high-resolution, coupled Earth System Model"

_The Cryosphere, 2022_

## Referee Comment (RC3)

**Review of *Southern Ocean polynyas and dense water formation in a high-resolution, coupled Earth System Model* by Jeon et al.**

This study explores the ability of the coupled model E3SM to simulate coastal polynyas and Dense Shelf Water (DSW) formation. Both features are misrepresented in most global models which is problematic as DSW formation in coastal polynyas is the precursor for Antarctic Bottom Water formation. The study makes use of E3SM-HR, a newly developed version of E3SM with a higher horizontal resolution. E3SM-HR can simulate coastal polynyas reasonably well, setting the model apart from other global coupled models. Biases occur in regions where polynyas occur next to fast ice which is not simulated by the model. The model forms open ocean polynyas which do occur in the Weddell Sea but are much less common than simulated in the model. The formation of open ocean polynyas is a prevalent problem in global models and results from a too weak stratification allowing excessive deep convection in winter. The study further examines the ability of E3SM-HR to form DSW on the continental shelf and finds that dense water is formed but it is not dense enough to form Antarctic Bottom Water. The reason for the water not being dense enough is due to a lack of onshore transport of salty Circumpolar Deep Water (CDW). The study states that a too strong Antarctic Slope Current driven by too strong winds resulting from a deep subpolar pressure system in E3SM-HR is the reason for the lack of onshore CDW transport and preconditioning of the water column.

The study is well structured and the figures are appropriate. The study presents a problem common to many models and the detailed description of the model performance regarding polynyas and dense water formation provides a beneficial contribution to the community.

My main comments can be summarised as follows:

1) Data selection (section 2.2 Ocean, sea ice, and atmosphere state estimates)
   a. Please explain the advantage and disadvantage of the SOSE and WOA data sets. Section 2.2 reads as if SOSE provides only surface values while WOA also provides subsurface data. The cross-slope transects of Fig 7-9 show SOSE data (=subsurface data). A short discussion of which data set is appropriate for the comparison would be helpful.
   b. What are the uncertainties of the observational datasets? Figure 2 shows uncertainties for the sea ice products, but nothing is presented for the oceanographic datasets. Differences between the model and observations/reanalysis products are also introduced by different temporal and spatial resolution which are not discussed. See specific comments below for suggestions.
2) The text is at times not clear about what results are drawn from figures presented in this study and what results are drawn from the literature. Please revise section 4.2 which discusses the different shelf regimes in Fig 7-9 comparing the model to SOSE and to observations from Whitworth et al. 1998.
3) The important finding of the study is that the subpolar low-pressure system around Antarctica is too low with consequences for the ASC, sea ice transport,

preconditioning of the water column, and DSW formation. Can you think of ways to test this hypothesis, or do you have ideas how to fix the problem in E3SM-HR? Adding a paragraph would be a valuable addition to the conclusions.

**Specific comments**

L1-2: The process described here is true for the formation of DSW. AABW forms when DSW escapes the continental shelf where it flows down the continental slope and entrains CDW and surface waters as described in L24-26. What is described here is only a part of the process. Can you adjust the sentence?

L45: What model biases? Please elaborate.

L50-54: Please specify if "coupled" refers to ocean-sea ice models or atmosphere-ocean-sea ice models. Can you comment on the performance of ocean-sea ice models where the atmosphere is prescribed?

L130: Is the definition for coastal polynyas the same in the model as in the observational data sets? You mention the definition you use to define coastal polynyas in E3SM, but not the definition for the observational data sets.

L145: "Cold katabatic winds blowing off Antarctica create sea ice" Please elaborate on the connection between katabatic winds, latent heat flux, and sea ice production. It might be helpful to add some references or explain in more detail how Fig 1 shows the connection between winds and sea ice production.

L147-148: Please be clear about what is directly shown in the figure and what are the conclusions or hypotheses you draw from the figure. High sea ice production rates are shown, but the generation of cold and saline water and the transport down the continental slope are an assumption (i.e. not shown in Fig 1b).

L185: Please clarify why Dufour et al. (2017) is referenced here. Did they use the same/different data? Or a model?

L189: "thus allowing for dense-water formation in E3SM-HR" → Is this an assumption or shown in Fig 3?

L200-203: Not shown in figure. Please be clear about what is shown on the figure and which statement is based on previous work and add reference accordingly.

L210-213: Do the models in the cited studies simulate coastal polynyas similar to E3SM-HR?

L225-226 and 227-228: How do the WMT rates compare to observed values or to models that are able to produce DSW?

L230-231: I am not sure I understand the sentence. 28kgm-3 is the density found in E3SM-HR as the threshold for AABW formation? And Orsi et al. (1999) find that it needs to be 28.28kgm-3?

L252-253 and L256-257: The text is not clear about what is new about the connection between wind and the ASC. Do the authors try to make the point that (i) wind is one of the drivers of the ASC (established knowledge) or that (ii) the strong winds in E3SM-HR are thought to be the reason for the strong ASC (this study)?

L271: The example transect for the dense regime at 35°W is at the eastern end of the section characterised by a dense shelf (e.g. Thompson et al. 2018). Why was this transect chosen? The dense shelf is better established in the western Weddell Sea.

L275ff: How was the exact transect location picked in the model output? The ASC varies substantially on small scales and comparing the observational transect with the simulated transect which is likely not exactly at the same location (limitations due to model grid) may introduce an error.

L287: Why chose this transect if SOSE does not show a dense shelf here? See comment L271.

L287-288: Does E3SM-HR show V-shaped isopycnals along the shelf break at all?

L289: Please comment on the fact that SOSE does not show a dense regime, but Whitworth et al. (1998) does. What does this mean for the comparison between E3SM-HR and SOSE to evaluate the model?

L308: Please see comment L304 regarding the poleward Ekman transport. The argumentation is based on enhanced Ekman transport but no evidence is presented.

L311: Please see general comment on section 2.2 regarding the choice of observational dataset used to compare the model to.

The discussion would benefit from an introductory sentence which states how the model compares to the chosen observational/reanalysis datasets.

L323-L330: The first paragraph of the discussion talks about one drawback of the reanalysis products, namely that they are not coupled. Is this the biggest uncertainty when comparing E3SM-HR with reanalysis data? If not, I suggest moving this paragraph further down.

L343-356: Can you comment on the importance of ice shelves for polynyas? Should they be added to the discussion of mechanisms that are not represented in E3SM?

L384-388: The paper finds that the deep subpolar low-pressure system is the reason for model biases. Can you think of ways to test this hypothesis or do you have ideas how to fix the problem? Adding a sentence or two would be a nice addition to the conclusions.

**Technical comments**

Abstract

L1: Rephrase to "of **the** Earth's".

L9: Delete "hence" or alternatively rewrite "and hence to too little".

L9: Replace "communication" with "exchange".

Introduction

L17: Rephrase to "areas of **ice-free surface water** or of thin, …".

L19: Rephrase to "advection of **sea** ice".

L21: Rephrase to "important role in **the** climate".

L22: Rephrase to "atmosphere **and thereby** affecting".

L23: I suggest changing the order "resulting from high rates of surface cooling and sea ice production". Sea ice production is a result of surface cooling and should be listed second.

L24: Replace "The latter" with "Point 2". It is not clear to me what "the latter" refers to (the entire second point or the sea ice formation only).

L27: Rephrase to "as **AABW is** an important sink".

L37: Rephrase to "are dense due to" or "are cold and salty due to". Highlighting that the water is *cold and dense* would only be informative when it is compared to a situation where the water is *warm and dense*.

L41: Add citation for onshore CDW transport, e.g., Stewart et al. 2015, Foppert et al. 2019.

L56: Rephrase to "Instead, **many** GCMs create AABW" to prevent repetition of the word *often* in the next sentence.

L57: Use the acronym GCM. Either always write the full name or always write the acronym. Please go through the entire manuscript and check for consistency for all acronyms.

L59: Rephrase to "this **vertical** heat transfer creates".

L60: Rephrase to "creating AABW by **surface heat loss**".

Data and methodology

L87: Can you specify the effect of the tuning parameters.

L91: Please always use the same order (e.g., first high-resolution, then low-resolution).

L94: Rephrase to "mesh has **a** resolution".

L95: Swap order of 30km and 60km. I assume 60km is at the equator and 30km at the poles? Please use the same order as in L92.

L95: Rephrase to "with **a** layer thickness".

L99: Remove hyphen between "1 and 3 months".

L105: Rephrase to "in **the** fully coupled simulations".

L120: Rephrase to "we use **the** SOSE data set".

L136: Please always use acronym (OOP) after introducing it, see comment on L57. Please check entire manuscript (e.g. L183).

L139: Rephrase to "These processes **occur** in areas of **an** ice-free ocean."

Antarctic coastal and open-ocean polynyas

L144: It is not clear to me why E3SMR-LR is in brackets. I suggest to simply rephrase to "simulated in **E3SM** by comparing".

L150-152: Rephrase to "In Fig 1c, we compare E3SM-HR's accumulated sae ice volume **in Antarctic coastal polynyas from March to October and as a function of longitude** with **the** satellite estimate **AMSR-E**."

L158: Remove "relatively".

L175: Rephrase to "**sum** of polynya area and sea ice volume production for polynyas **that are not** associated with landfast ice".

L183: Remove comma after "large".

L186. Please split sentence into two "…at any point in the simulation**.** E3SMR-HR does produce MRPs…".

L194: Rephrase to "the strong **sub**polar cyclonic gyre".

L195: Rephrase to "preconditioning this convect**ive** process".

L195: Rephrase to "pycnocline and **a** circulation".

L199: Rephrase to "or **in which** embayment-like feature".

Dense water formation

L214: Rephrase to "last 30 years of **the** E3SM-HR simulation".

L221: Rephrase to "produced in E3SM-HR **for** all surface fluxes combined".

L225: Specify the regions where SWMT is negative. Are they relevant?

L230: Suggest splitting the sentence after "AABW formation".

L233-234: Rephrase to "similarly to **what has been found in previous** studies".

L260: The low salinity on the shelf in the fresh regime is a result of the winds and onshore Ekman transport. Can you change the order of the sentence to distinguish between mechanism and the resulting stratification?

L263-267: Add information that the dense shelf is important for onshore CDW transport and preconditioning of shelf waters for DSW formation.

L273-274: The sentence reads as if the data from Whitworth et al. (1998) is shown in Fig 7-9, but that is not the case. Please clarify.

L282: Rephrase to "**formed** by brine rejection".

L283: Rephrase to "**isopycnals associated with** warm CDW **tilt** down towards the seafloor".

L284: Rephrase to "**the isopycnals shoal** again".

L291: Check that the reference to the figure is correct. It should be referring to Fig 8 here.

L295-297: Are the first three sentences referring to information from Thompson et al. (2018) or are they based on the model simulation?

L298: Rephrase to "SOSE has **isohaline surfaces that tilt upward towards** Antarctica".

L304: This sentence focuses on the strong zonal winds and assumes an effect on onshore Ekman transport. But the meridional winds are also larger than in ERA5. Did you look at the Ekman transport in the model and its zonal and meridional components?

L307: Rephrase to "build-up".

L311: Remove "same".

Discussion

L327: Replace "ingest" with "process".

L331: Delete "whereas".

L331: E3SM-HR has an atmospheric model component, what is the meaning of "atmospheric forcing" (HighResMIP) here? My understanding from reading section 2.2 is the output of the model simulation is used in HighResMIP. Please clarify.

L332: Split the sentence into two: "… of a stable climate**. T**he observations with which **E3SM-HR is compared to** are for a transient climate."

L323: Remove "moreover".

L336: Remove "random differences in".

Summary and conclusions

L358-359: Also mention E3SM-LR.

L366: Rephrase to "Aside from the large **embayment-shaped polynyas**, …".

L368: Rephrase to "which are **almost** entirely due to".

L273: Overly strong southward Ekman transport not shown in study.

L376: Rephrase to "there is no **DSW** formation".

L379: Rephrase to "thick sea ice in **s**ummer".

Tables

Table 1: What are the atmosphere and land listed twice? What does 72 and 15 mean?
Table 2: Rephrase to "state **estimate** datasets" and add full stop.
Table 3: Add full stop.

Figures

Figure 1

- Panel a) Why is there green shading on land? The latent heat flux from the ocean to land should only have values over the ocean.
- Figure caption: Rephrase to "(d) Total sea ice volume production **per** year".

Figure 2
- Please use the same colour (pink) for E3SM-HR as in Figure 1.
- Figure caption: Rephrase to "(a-m) **Area-volume** diagrams"; "(n) **Integrated** polynya area"; "(o) **Integrated** polynya area".

Figure 3
- Figure caption: Rephrase to "(c) November from NCDR **(year** 2017)"; (i) November **(model year 54)**.

Figure 4
- Show pink box indicating the Weddell Sea region in panel (a) only and adjust the figure caption accordingly: "The pink **box in (a) indicates** the area…"

Figure 5:
- Panel (a): Is the WMT rate shown on the map the integrated WMT rate over all density classes?
- Figure caption: Add information that the figure is from E3SM-HR output.
- "Note that the green and pink curves do not sum to the gray curve." What is missing?

Figure 6
- Colorbars should start at zero and not have an arrow on the left end. Zero is the smallest possible value.

Figure 7-9
- Discuss the impact of the fact that the SOSE and E3SM-HR transects are not at the exact same position.

Figure 10:
- The diverging colormap is confusing, please use a sequential colormap.
- Colorbar should start at zero and not have an arrow, see comment on Figure 6.
- Showing a difference plot (SOSE-E3SMR-HR) would help identifying regions where sea ice thickness differs.

Figure 11:
- See last comment on Figure 10. Difference plots make it easier to see where the models differ from observations.
- Please use sequential colormap for sea surface neutral density.

**New references**

Stewart, A. L., and A. F. Thompson (2015), Eddy-mediated transport of warm Circumpolar Deep Water across the Antarctic Shelf Break, Geophys. Res. Lett., 42, 432–440, doi:10.1002/2014GL062281.

Foppert, A., Rintoul, S. R., & England, M. H. (2019). Along-slope variability of cross-slope eddy transport in East Antarctica. Geophysical Research Letters, 46, 8224–8233. https://doi.org/10.1029/ 2019GL082999
Received

---

## Author Comment (AC1)

**Response to Reviewer 1 for "Southern Ocean polynyas and dense water formation in a high-resolution, coupled Earth System Model" by Jeong et al.**

We thank the reviewer for their helpful and constructive comments. Please find our responses below (the reviewer's comments are noted in italics and our reference to manuscript line numbers refers to the revised version of the manuscript).

*... The study by Jeong et al. contributes to the study of coastal polynyas and the formation of dense/bottom water in GCMs... The manuscript is well-written and includes high-quality figures to support the main results. The methodology is sound and builds on state-of-the-art coupled Earth System Model (ESM) development. However, the discussion section is too brief and offers limited new insight into the main topic of missing dense-water formation in the coastal polynyas in GCMs and ESMs. I summarize key aspects the authors should address to improve the manuscript and its contribution to the ESM development near the Antarctic shelves:*

*1. Increasing the resolution of the E3SM improved the representation of coastal polynyas, but the associated dense-water formation was too weak and did not produce dense water with similar characteristics as the observations. What is your advice to the ESM community to further improve the polynya representation?*

We believe that, while high horizontal resolution facilitates the formation of coastal polynyas because of the better resolved katabatic winds and ocean circulation near the coast, a proper cross-shelf ocean circulation also needs to be simulated. And that in turn is driven by well reproduced alongshore winds. We have now added a paragraph discussing this, as well as other points raised by the reviewer below, in the Discussion section: see text in blue starting from line

326.

*2. You show that the winds over the continental shelves are too strong, resulting in a too strong Antarctic Slope Front which prevents shelf-ocean water exchange. What means can you do to get a better representation of the winds over the Antarctic shelves? Would it be possible to run the E3SM-HR with an idealized wind forcing near Antarctica to test the response to strong and weak wind scenarios? Including such a test in the manuscript would strongly increase the relevance of this study to the modeling community.*

We thank the reviewer for these suggestions. Although we did not have the resources (including additional dedicated computational time) to run new E3SM experiments at the time of these revisions, we have carefully re-analyzed the E3SM-HR simulation by selecting years with strong and weak winds at the dense-shelf location ($63°$S). Specifically, the strong-wind (weak-wind) years were selected as years in which the wind speed for the winter season were higher (lower) than the top (bottom) 25% of the 30-year wind speeds. We were then able to compute strong-wind and weak-wind composites for cross-shelf ocean temperature, salinity, and alongshore velocity, and have included the results in a new figure, current Figure 8. These results support our hypothesis that the strong winds are an important contributor to the overly strong Antarctic Slope Front in E3SM-HR. We have added a discussion about this starting on line 288.

*3. You describe plans to include landfast ice to improve the model further. How will the representation of landfast ice improve the model if the winds remain unchanged?*

Landfast sea ice contributes to coastal polynya formation by effectively changing the coastline. This process, which is independent from the wind driving mechanism, is now briefly discussed in

section 5, line 332.

*4. The Southern Ocean is an area of upwelling. How well is the E3SM-HR representing the water mass characteristics in the Southern Ocean, and how are these characteristics affecting the formation of open-ocean polynyas?*

We thank the reviewer for this question. We have now included a new figure in the Supplementary Material (Supplementary Fig. 1) showing T/S diagrams from E3SM-HR and WOA18. We have discussed these results on lines 235-239. Furthermore, we have added a sentence clarifying that the likely reason for OOPs formation in E3SM-HR is the concurrency of a strong Weddell Gyre and a weaker stratification (results not shown) in the interior Weddell Sea (lines 196-199).

*5. The E3SM-HR does not include ice shelf cavities where dense water interacts with the ice shelf base to form the densest versions of shelf water (Ice shelf water). How are the missing ice shelves and the presence of ISW affecting the model performance?*

We have added a sentence including the possible effects of ice shelves in the Discussion section (line 330).

*6. The modeled sea ice is much thinner than the observations in OOP years over the whole WS. How is this affecting the dense water formation and the likelihood of increasing the lifespan of an OOP?*

We assume that the reviewer is referring to Fig. 3a-c compared to Fig. 3d-f (please let us know if that is not the case). In that figure, we intend to use the observations only as reference, and not for model evaluation purposes. This is because OOP's are intermittent and highly variable events, and we cannot compare model and observation single events directly. We have added a sentence

to clarify this in reference to Fig. 3a-c (see text in blue on line 179). Having said that, the question of how a less sea ice covered Weddell Sea would impact the lifespan of an OOP is an interesting one that deserves investigating in a separate study.

*7. Are there any caveats in comparing the model with the SOSE database? Could it make sense to also compare the transects in Figures 7-9 with CTD sections?*

We followed the reviewer's suggestion and replaced SOSE with CTD observations for the continental shelf sections in Figure 7. These are the same in-situ data used in Thompson et al. (2018).

***Minor comments:***

*Line 159: It is very hard to see the "relatively higher latent heat flux...in East-Antarctica" in Fig1a.*

We have now modified the color palette and masked out Antarctica in Figure 1a.

*Line 256: It is very hard to see the easterly winds in figure 6b.*

We have now updated Figure 6b, making the wind vectors more visible.

---

## Author Comment (AC2)

**Response to Reviewer 2 for "Southern Ocean polynyas and dense water formation in a high-resolution, coupled Earth System Model" by Jeong et al.**

We thank the reviewer for their helpful and constructive comments. Please find our responses below (the reviewer's comments are noted in italics and our reference to manuscript line numbers refers to the revised version of the manuscript).

*The authors investigate dense water and polynya formation in two versions of the Energy Exascale Earth System Model. The manuscript is clearly structured and the results are well presented. Some of the arguments don't seem to be supported by the material and the results need to be expanded upon.*

*Main comments here:*

*1. Could you give recommendations for development? What are the key improvements needed here (easterlies and Ekman transport)?*

Thank you for this suggestion. We have now added a paragraph discussing these points in section 5: see text in blue starting from line 326.

*2. You mention the impact of the overly strong polar easterlies and associated Ekman transport throughout the paper and abstract. However - you do not seem to directly calculate the Ekman transport. Can you do this and compare to observations?*

We have followed the reviewer's suggestion and calculated the Ekman transport for the E3SM-HR simulation and the ERA5 data. The results are shown in Figure 3 of the Supplementary Material and discussed on line 297.

*3. The mean-state open ocean stratification is important for the formation of open ocean polynya in models. What does the open ocean stratification look like in these models compared to observations? Is it overly weak (explaining the convection in HR model) or overly strong (explaining no convection in the LR model)?*

We thank the reviewer for this question. Indeed, compared with SOSE, the upper ocean stratification in the interior Weddell Sea is too strong in E3SM-HR and too weak in E3SM-LR. We have not included results showing this in the paper but i) we have added a discussion mentioning this on lines 196-199 and 206-208, and ii) we have included potential density results for E3SM-HR and E3SM-LR at the surface and at 200 m depth in Figs. 1, 2 below, for the reviewer's convenience.

*4. You mention that the model is fully coupled. The ocean-ice interaction is very important for DSW and presumably the coastal polynya development. Please include a discussion of this.*

We have explicitly mentioned air-sea interaction processes in various parts of the manuscript. Please see, for example, lines 145, 184, 241-244.

***Minor comments:***

*Line 18: Are you referring to katabatic winds? If so, please introduce term as you use it again later.*

Thank you from bringing this up. We now use the term "katabic winds" throughout the manuscript.

*Line 20: Are coastal polynya important for other aspects of the earth system e.g., marine biology or biogeochemical cycles?*

Indeed. We have now added a sentence on line 22.

*Line 20: You may introduce that these are areas of high sea ice production. Also, coastal polynya may form due to oceanic currents.*

Done. Please see line 17.

*Line 46: Could add citation to On the Role of the Antarctic Slope Front on the Occurrence of the Weddell Sea Polynya under Climate Change.*

Done.

*Line 55: You may introduce the two types (coastal and open) of polynya in paragraph one.*

Done. Please see lines 26-29.

*Line 134: You can have open ocean convection that doesn't form polynya in models (e.g., Dufour et al 2017 and Lockwood et al 2020).*

We are not entirely sure what sentence the reviewer is referring to. If this is the sentence: "These polynyas are generally observed in conjunction with deep convection events in the ocean..." (now on line 28), then we do not think that this implies that whenever there is deep convection a polynya forms. But rather that deep convection is always found in OOPs.

*Line 181: Moved westward with an average velocity of $0.013\,\mathrm{m\,s}^{-1}$ (Gordon 1978, 1982).*

Thank you for the clarification. Revised text is now on line 175.

*Line 193: Episodic open ocean deep convection events in the Weddell Sea have been linked to anomalies in the Southern Annular Mode index (Gordon et al. 2007; Cheon et al. 2014; Francis et*

*al. 2019; Campbell et al. 2019; Cheon and Gordon 2019). Have you considered the representation the SAM in these models?*

We have now added a sentence describing the studies mentioned by the reviewer in the revised version of the manuscript, lines 191-192.

*Line 209: You can have convection and dense water formation without polynya formation (see Dufour et al. 2015 and Lockwood et al. 2020). Please check if the LR model is in fact creating dense water and convection, just without polynya formation. Convection can be calculated via. the mixed layer depth (see de Lavenge et al. 2015).*

In addition to Fig. 2, we have included Fig. 3 (maximum mixed layer depth) below from both E3SM-HR and E3SM-LR. Clearly, there is no evidence of deep convection in the low-resolution simulation. We have added a sentence clarifying this on lines 206-208 of the revised manuscript.

*Lines 258: Although you're correctly taking the transects from the respective areas following Thompson et al 2018 - the Western and Eastern Weddell Sea transects seem very close together. I'd like to see how this model holds in another region of Fresh Shelf (e.g., along the Ross sea). Also, can the LR model capture the hydrography?*

As the reviewer suggested, we re-selected the three transects for representing three types of shelves, and now they are at the same location as those considered in Thompson et al. (2018). We also compared the vertical temperature and density fields of the E3SM-HR simulation with CTD observations rather than SOSE. The revised vertical cross-sections are shown in Figure 7.

*Line 295: Could you produce depth average velocities around the full Antarctic?*

Please see the depth-averaged oceanic currents speed for E3SM-HR and SOSE in Fig. 4 below. As expected, the model currents are stronger along the Antarctic coast with respect to SOSE.

*Line 325: Have you considered comparing to very high-resolution simulations like the MITGCM or LCM-4320?*

Thank you for this suggestion. However, we are unable to compare with those simulations at the moment. Instead, we compared our E3SM-HR results with CTD observations in Figure 7.

*Figure 6. c-d The vectors are difficult to see in the figures.*

Thank you for the suggestion. We modified the vectors to make them more visible.

**LIST OF FIGURES**

[Figure]

FIG. 1. Potential density ($\sigma_0$) from E3SM-HR (left) and SOSE (middle) at the surface (upper) and 200 m depth (lower). Right panels show model-SOSE biases.

[Figure]

FIG. 2. Similar to Fig. 1 but for E3SM-LR.

[Figure]

FIG. 3. Maximum depth of the mixed layer (m) in any month of the 30 simulation years for (a) E3SM-HR and (b) E3SM-LR.

**Annual-mean depth averaged velocities**

[Figure]

F<small>IG</small>. 4. Annual-mean depth-averaged oceanic currents speed for (a) SOSE and (b) E3SM-HR.

---

## Author Comment (AC3)

**Response to Reviewer 3 for "Southern Ocean polynyas and dense water formation in a high-resolution, coupled Earth System Model" by Jeong et al.**

We thank Reviewer 3 for their helpful and constructive comments. In order to address the reviewer's central concern - separating text results drawn from figures presented in this study and results drawn from the literature - we have revised the manuscript considerably. Also, we included CTD observational data for validating the model cross-shelf sections. These results are now shown in Figure 7. Please find our responses to each point raised by the reviewer below (the reviewer's comments are noted in italics and our reference to manuscript line numbers refers to the revised version of the manuscript).

*This study explores the ability of the coupled model E3SM to simulate coastal polynyas and Dense Shelf Water (DSW) formation... The study is well structured and the figures are appropriate. The study presents a problem common to many models and the detailed description of the model performance regarding polynyas and dense water formation provide a beneficial contribution to the community.*

***My main comments can be summarised as follows:***

*1. Data selection (section 2.2 Ocean, sea ice, and atmosphere state estimates)*

*a. Please explain the advantage and disadvantages of the SOSE and WOA data sets. Section 2.2 reads as if SOSE provides only surface values while WOA also provides subsurface data. The cross-slope transects of Fig 7-9 show SOSE data (=subsurface data). A short discussion of which data set is appropriate for the comparison would be helpful.*

Thank you for this suggestion. We have now added a short discussion in Section 2.2 (please see lines 120-128) as well as included CTD in-situ observational data to better evaluate the model for

the cross-shelf sections (new Figure 7).

*b. What are the uncertainties of the observational datasets? Figure 2 shows uncertainties for the sea ice products, but nothing is presented for the oceanographic datasets. Differences between the model and observations/reanalysis products are also introduced by different temporal and spatial resolution which are not discussed. See specific comments below for suggestions.*

We agree with the reviewer on these points. We have now added the following text mentioning potential caveats with quantifying uncertainties in the discussion section (please see lines 345-347): "Different temporal and spatial resolutions also introduce discrepancies between observations/reanalysis products and model results. A potential caveat to note is that we only used one high-resolution and one low-resolution model simulation, thus not providing a quantification of structural model uncertainty."

*2. The text is at times not clear about what results are drawn from figures presented in this study and what results are drawn from the literature. Please revise section 4.2 which discusses the different shelf regimes in Fig 7-9 comparing the model to SOSE and to observations from Whitworth et al. 1998.*

We would like to thank the reviewer for this helpful comment. We have now substantially changed section 4.2, clarifying the points raised by the reviewer and also providing additional evidence for our main conclusions. Please refer to revised section 4.2 (text in blue contains the main changes).

**Specific comments**

*L1-2: The process described here is true for the formation of DSW. AABW forms when DSW escapes the continental shelf where it flows down the continental slope and entrains CDW and*

*surface waters as described in L24-26. What is described here is only a part of the process. Can you adjust the sentence?*

Agreed. We have modified that sentence as follows: "Antarctic coastal polynyas produce dense shelf water, a primary source of Antarctic Bottom Water that maintains global overturning circulation."

*L45: What model biases? Please elaborate.*

We have removed "and/or model bias" from that sentence after carefully reviewing Dinniman et al. 2016 and Thompson et al. 2018.

*L50-54: Please specify if "coupled" refers to ocean-sea ice models or atmosphere-ocean-sea ice models. Can you comment on the performance of ocean-sea ice models where the atmosphere is prescribed?*

The "coupled" refers to ocean-sea ice models. We have changed the sentence to "coupled ocean-sea ice models with prescribed atmosphere" (line 51). We also comment on the performance of ocean-sea ice models on lines 54-55 as follows: "However, these models tend to simulate larger coastal ice production due to the lack of sensible and latent heat flux transferred from the ocean to the atmosphere."

*L130: Is the definition for coastal polynyas the same in the model as in the observational data sets? You mention the definition you use to define coastal polynyas in E3SM, but not the definition for the observational data sets.*

We have now clearly stated how the coastal polynyas are defined in the model and in the observations: please see text in blue on lines 136-138.

*L145: "Cold katabatic winds blowing off Antarctica create sea ice" Please elaborate on the connection between katabatic winds, latent heat flux, and sea ice production. It might be helpful to add some references or explain in more detail how Fig 1 shows the connection between winds and sea ice production.*

Thank you for this suggestion. We have now added more detail as suggested on lines 144-146, as follows: "Cold katabatic winds blowing off Antarctica in E3SM-HR (Fig. 1a) push the thick sea ice offshore, leading to open water. Consequently, intensive latent heat fluxes transfer from the ocean to the atmosphere makes favorable conditions for sea ice production in the Antarctic coastal polynyas (Fig. 1b)."

*L147-148: Please be clear about what is directly shown in the figure and what are the conclusions or hypotheses you draw from the figure. High sea ice production rates are shown, but the generation of cold and saline water and the transport down the continental slope are an assumption (i.e. not shown in Fig 1b).*

Apologies about this. The sentences that are not related to the description of the figure have now been removed.

*L185: Please clarify why Dufour et al. (2017) is referenced here. Did they use the same/different data? Or a model?*

We have now modified the sentence (line 181) as follows: "Similar to a previous study comparing the difference between low and high-resolution climate model results (e.g.; Dufour et al. 2017),

E3SM-LR does not exhibit OOPs at any point in the simulation."

*L189: "thus allowing for dense-water formation in E3SM-HR" → Is this an assumption or shown in Fig 3?*

Thank you for pointing this out. We modified the sentence on line 185 as follows: "which can potentially affect the dense-water formation in E3SM-HR."

*L200-203: Not shown in figure. Please be clear about what is shown on the figure and which statement is based on previous work and add reference accordingly.*

Apologies for the confusion. We have now reorganized the text of the two paragraphs on lines 190-199 and 200-204, adding clear reference to previous work and to our Figure 4.

*L210-213: Do the models in the cited studies simulate coastal polynyas similar to E3SM-HR?*

We have now removed that sentence.

*L225-226 and 227-228: How do the WMT rates compare to observed values or to models that are able to produce DSW?*

Thank you for raising this question. We have now compared the model results to the observational study of Pellichero et al. 2018 (please see lines 224 and 227).

*L230-231: I am not sure I understand the sentence. 28kgm-3 is the density found in E3SMHR as the threshold for AABW formation? And Orsi et al. (1999) find that it needs to be 28.28kgm-3?*

Apologies for the confusion. We modified the sentence (see lines 227-230) as follows. "In the high density ranges above $28.0 \, \mathrm{kg\,m^{-3}}$, a local maximum WMT rate of $3.0 \, \mathrm{Sv}$ is seen at a neutral

density of $28.2\,\mathrm{kg\,m^{-3}}$. Whitworth et al. (1998) and Orsi et al. (1999) define AABW as having neutral densities higher than $28.27\,\mathrm{kg\,m^{-3}}$. We consider $28.0\,\mathrm{kg\,m^{-3}}$ to be a suitable minimum threshold of AABW in the E3SM-HR simulation."

*L252-253 and L256-257: The text is not clear about what is new about the connection between wind and the ASC. Do the authors try to make the point that (i) wind is one of the drivers of the ASC (established knowledge) or that (ii) the strong winds in E3SM-HR are thought to be the reason for the strong ASC (this study)?*

We have now clarified the text on lines 258-259 as follows: "Therefore, we hypothesize that the strong winds over the continental shelf may be one of the drivers of the strong ASC in E3SM-HR."

*L271: The example transect for the dense regime at 35°W is at the eastern end of the section characterized by a dense shelf (e.g. Thompson et al. 2018). Why was this transect chosen? The dense shelf is better established in the western Weddell Sea.*

Thank you for pointing this out. We re-selected the position of the dense shelf as shown in Thompson et al. (2018). Please refer to Figure 6d for section location.

*L275: How was the exact transect location picked in the model output? The ASC varies substantially on small scales and comparing the observational transect with the simulated transect which is likely not exactly at the same location (limitations due to model grid) may introduce an error.*

We have now re-selected the transect locations to coincide with the cross-sections shown in Thompson et al. 2018, where CTD observational data is also available for comparison with the model results. Please refer to the revised Figure 7.

*L287: Why chose this transect if SOSE does not show a dense shelf here? See comment L271.*

As mentioned above, we no longer use SOSE for the vertical cross-section plots.

*L287-288: Does E3SM-HR show V-shaped isopycnals along the shelf break at all?*

We have looked at a number of sections in the Weddell as well as Ross Sea and we were not able to see V-shaped isopycnals using the 30-year average climatological fields.

*L289: Please comment on the fact that SOSE does not show a dense regime, but Whitworth et al. (1998) does. What does this mean for the comparison between E3SM-HR and SOSE to evaluate the model?*

We no longer use SOSE for the vertical cross-section plots.

*L308: Please see comment L304 regarding the poleward Ekman transport. The argumentation is based on enhanced Ekman transport but no evidence is presented.*

Thank you for pointing this out. We have now added a plot of poleward Ekman transport computed from the E3SM-HR simulation and from ERA5 fields in the Supplementary Material.

*L311: Please see general comment on section 2.2 regarding the choice of observational dataset used to compare the model to. The discussion would benefit from an introductory sentence which states how the model compares to the chosen observational/reanalysis datasets.*

We updated Section 2.2 and clarified which dataset is used for each comparison.

*L323-L330: The first paragraph of the discussion talks about one drawback of the reanalysis products, namely that they are not coupled. Is this the biggest uncertainty when comparing E3SM-HR with reanalysis data? If not, I suggest moving this paragraph further down.*

We have followed the reviewer's suggestion and moved this paragraph further down.

*L343-356: Can you comment on the importance of ice shelves for polynyas? Should they be added to the discussion of mechanisms that are not represented in E3SM?*

We have now added this sentence on lines 330-332: "Another important but missing piece of E3SM-HR is the representation of ice shelves, which closely interact with coastal polynya and DSW (Jeong et al. 2020)."

*L384-388: The paper finds that the deep subpolar low-pressure system is the reason for model biases. Can you think of ways to test this hypothesis or do you have ideas how to fix the problem? Adding a sentence or two would be a nice addition to the conclusions.*

We have done additional analysis and presented more evidence for our hypothesis in the revised version of the manuscript. Please see Figure 7 and related discussion, and the Ekman transport figure in Supplementary Material.

***Technical comments***

We sincerely appreciate Reviewer 3 for taking the time to read our manuscript in detail. We have followed all the reviewer's suggestions, except for those cases where the text had been modified to the point that the suggestion was no longer applicable.

***Abstract***

*L1: Rephrase to "of the Earth's".*

Corrected (now on line 25).

*L9: Delete "hence" or alternatively rewrite "and hence to too little".*

Corrected.

*L9: Replace "communication" with "exchange".*

Corrected.

**Introduction**

*L17: Rephrase to "areas of ice-free surface water or of thin, . . . ".*

Corrected.

*L19: Rephrase to "advection of sea ice".*

Corrected.

*L21: Rephrase to "important role in the climate".*

Corrected.

*L22: Rephrase to "atmosphere and thereby affecting".*

Corrected.

*L23: I suggest changing the order "resulting from high rates of surface cooling and sea ice production". Sea ice production is a result of surface cooling and should be listed second.*

Done.

*L24: Replace "The latter" with "Point 2". It is not clear to me what "the latter" refers to (the entire second point or the sea ice formation only).*

Corrected.

*L27: Rephrase to "as AABW is an important sink".*

Done.

*L37: Rephrase to "are dense due to" or "are cold and salty due to". Highlighting that the water is cold and dense would only be informative when it is compared to a situation where the water is warm and dense.*

Done.

*L41: Add citation for onshore CDW transport, e.g., Stewart et al. 2015, Foppert et al. 2019.*

Thank you. References added.

*L56: Rephrase to "Instead, many GCMs create AABW" to prevent repetition of the word often in the next sentence.*

Corrected.

*L57: Use the acronym GCM. Either always write the full name or always write the acronym. Please go through he entire manuscript and check for consistency for all acronyms.*

Done.

*L59: Rephrase to "this vertical heat transfer creates".*

Corrected.

*L60: Rephrase to "creating AABW by surface heat loss".*

Corrected.

***Data and methodology***

*L87: Can you specify the effect of the tuning parameters.*

We feel that this is too much information for this paper. Instead, we are referring to the Caldwell et al. (2019) paper.

*L91: Please always use the same order (e.g., first high-resolution, then low-resolution).*

Done.

*L94: Rephrase to "mesh has a resolution".*

Corrected.

*L95: Swap order of 30km and 60km. I assume 60km is at the equator and 30km at the poles? Please use the same order as in L92.*

No, the current text is correct: 60 km resolution at the mid-latitudes and 30 km at the equator and poles.

*L95: Rephrase to "with a layer thickness".*

Corrected.

*L99: Remove hyphen between "1 and 3 months".*

Corrected.

*L105: Rephrase to "in the fully coupled simulations".*

Corrected.

*L120: Rephrase to "we use the SOSE data set".*

Done.

*L136: Please always use acronym (OOP) after introducing it, see comment on L57. Please check entire manuscript (e.g. L183).*

Done.

*L139: Rephrase to "These processes occur in areas of an ice-free ocean."*

Corrected.

*L144: It is not clear to me why E3SMR-LR is in brackets. I suggest to simply rephrase to "simulated in E3SM by comparing".*

Corrected.

*L150-152: Rephrase to "In Fig 1c, we compare E3SM-HR's accumulated sea ice volume in Antarctic coastal polynyas from March to October and as a function of longitude with the satellite estimate AMSR-E."*

Done.

*L158: Remove "relatively".*

Corrected.

*L175: Rephrase to "sum of polynya area and sea ice volume production for polynyas that are not associated with landfast ice".*

Corrected.

*L183: Remove comma after "large".*

Corrected.

*L186. Please split sentence into two "...at any point in the simulation. E3SMR-HR does produce MRPs...".*

Done.

*L194: Rephrase to "the strong subpolar cyclonic gyre".*

Corrected.

*L195: Rephrase to "preconditioning this convective process".*

Corrected.

*L195: Rephrase to "pycnocline and a circulation".*

Corrected.

*L199: Rephrase to "or in which embayment-like feature".*

Done.

***Dense water formation***

*L214: Rephrase to "last 30 years of the E3SM-HR simulation".*

Corrected.

*L221: Rephrase to "produced in E3SM-HR for all surface fluxes combined".*

Corrected.

*L225: Specify the regions where WMT is negative. Are they relevant?*

Corrected.

*L230: Suggest splitting the sentence after "AABW formation".*

Done.

*L233-234: Rephrase to "similarly to what has been found in previous studies".*

Corrected.

*L260: The low salinity on the shelf in the fresh regime is a result of the winds and onshore Ekman transport. Can you change the order of the sentence to distinguish between mechanism and the resulting stratification?*

Done.

*L263-267: Add information that the dense shelf is important for onshore CDW transport and preconditioning of shelf waters for DSW formation.*

Done (see text in blue on lines 267-268).

*L273-274: The sentence reads as if the data from Whitworth et al. (1998) is shown in Fig 7-9, but that is not the case. Please clarify.*

The reference has now been removed.

*L282: Rephrase to "formed by brine rejection".*

Corrected.

*L283: Rephrase to "isopycnals associated with warm CDW tilt down towards the seafloor".*

Corrected.

*L284: Rephrase to "the isopycnals shoal again".*

Done.

*L291: Check that the reference to the figure is correct. It should be referring to Fig 8 here.*

Figure numbers have been double checked and corrected.

*L295-297: Are the first three sentences referring to information from Thompson et al. (2018) or are they based on the model simulation?*

Thank you for pointing this out. The first three sentences refer to information from Thopmson et al. (2018), which is now cited.

*L298: Rephrase to "SOSE has isohaline surfaces that tilt upward towards Antarctica".*

SOSE has now been replaced with CTD observational data in the revised Figure 7.

*L304: This sentence focuses on the strong zonal winds and assumes an effect on onshore Ekman transport. But the meridional winds are also larger than in ERA5. Did you look at the Ekman transport in the model and its zonal and meridional components?*

Thank you for this suggestion. We have now included a figure of Ekman transport in the Supplementary Material and added a discussion on lines 297-298.

*L307: Rephrase to "build-up".*

Corrected.

*L311: Remove "same".*

Done.

***Discussion***

*L327: Replace "ingest" with "process".*

Corrected.

*L331: Delete "whereas".*

Corrected.

*L331: E3SM-HR has an atmospheric model component, what is the meaning of "atmospheric forcing" (HighResMIP) here? My understanding from reading section 2.2 is the output of the model simulation is used in HighResMIP. Please clarify.*

We apologize for the confusion. This is meant as "atmospheric greenhouse gas forcing". We have clarified the sentence on lines 348-349: "The E3SM simulations presented in this paper use the fixed 1950s atmospheric forcing, which consists of greenhouse gases, O3, and aerosol for a 1950s (~10-year mean) climatology."

*L332: Split the sentence into two: "... of a stable climate. The observations with which E3SM-HR is compared to are for a transient climate."*

Done.

*L323: Remove "moreover".*

Corrected.

*L336: Remove "random differences in".*

Corrected.

***Summary and conclusions***

*L358-359: Also mention E3SM-LR.*

Done.

*L366: Rephrase to "Aside from the large embayment-shaped polynyas, . . . ".*

Corrected.

*L368: Rephrase to "which are almost entirely due to".*

Corrected.

*L273: Overly strong southward Ekman transport not shown in study.*

We now included a figure of Ekman transport computed from E3SM-HR and from the ERA5 data in the Supplementary Material.

*L376: Rephrase to "there is no DSW formation".*

Corrected.

*L379: Rephrase to "thick sea ice in summer".*

Corrected.

*Tables*

*Table 1: What are the atmosphere and land listed twice? What does 72 and 15 mean?*

We have now clarified the fields listed in Table 1.

*Table 2: Rephrase to "state estimate datasets" and add full stop.*

Done.

*Table 3: Add full stop.*

Corrected.

*Figures*

*Figure 1 Panel a): Why is there green shading on land? The latent heat flux from the ocean to land should only have values over the ocean.*

Corrected.

*Figure 1 caption: Rephrase to "(d) Total sea ice volume production per year".*

Done.

*Figure 2: Please use the same color (pink) for E3SM-HR as in Figure 1.*

Done.

*Figure 2 caption: Rephrase to "(a-m) Area-volume diagrams"; "(n) Integrated polynya area"; "(o) Integrated polynya area".*

Corrected. Thank you.

*Figure 3 caption: Rephrase to "(c) November from NCDR (year 2017)"; (i) November (model year 54).*

Corrected.

*Figure 4: Show pink box indicating the Weddell Sea region in panel (a) only and adjust the figure caption accordingly: "The pink box in (a) indicates the area..."*

Corrected.

*Figure 5 Panel (a): Is the WMT rate shown on the map the integrated WMT rate over all density classes?*

Yes, it is. We have now clarified this in the figure caption.

*Figure 5 caption: Add information that the figure is from E3SM-HR output.*

Done.

*Figure 5: "Note that the green and pink curves do not sum to the gray curve." What is missing?*

The gray curve is the summation of the WMT rate over the whole Southern Ocean (south of $60°$S). The pink curve indicates the WMT rate over the Weddell Sea and the green curve denotes the WMT rate over the Antarctic continental shelves as indicated in Figure 5a. We also notify that in the caption of Figure 5 as follows "Note that the green and pink curves do not sum to the gray curve."

*Figure 6: Colorbars should start at zero and not have an arrow on the left end. Zero is the smallest possible value.*

Thanks for the suggestions. We have modified the color bar.

*Figure 7-9: Discuss the impact of the fact that the SOSE and E3SM-HR transects are not at the exact same position.*

This comment is no longer relevant because SOSE has now been replaced with CTD observations in Figure 7.

*Figure 10: The diverging colormap is confusing, please use a sequential colormap.*

Done.

*Figure 10: Colorbar should start at zero and not have an arrow, see comment on Figure 6.*

Done.

*Figure 10: Showing a difference plot (SOSE-E3SMR-HR) would help identifying regions where sea ice thickness differs.*

Thank you for this suggestion. Unfortunately, the resolutions of SOSE and E3SM-HR are different from each other, therefore, we could not make the difference plot.

*Figure 11: See last comment on Figure 10. Difference plots make it easier to see where the models differ from observations.*

See response above.

*Figure 11: Please use sequential colormap for sea surface neutral density.*

Done.

---

## Referee Report (RR1)

**Review, round 2, of *Southern Ocean polynyas and dense water formation in a high-resolution, coupled Earth System Model* by Jeong et al.**

I thank the authors for thoroughly addressing each comment from my previous review. I only have very few minor comments aimed to further improve the manuscript. I recommend accepting the manuscript for publication after the minor comments have been addressed.

Minor comments

L51-57: Solodoch et al. 2022 analyse simulated AABW formation in a global ocean-sea ice model. The study should be mentioned in the introduction.

L223: Please elaborate on the comparison between the model and Pellichero et al. 2018: Do the results 'compare well' in magnitude, pattern or both?

L279: Please use ASF acronym.

Section 2.1 and discussion (L325): Regarding the horizontal resolution of E3SM-HR and resolving the mesoscale. Multiple modelling studies have shown that a resolution of 1-2 km is required to adequately simulate mesoscale eddy activity over the Antarctic continental shelf and slope (e.g., Nost et al. 2011, Dinniman et al. 2012, St-Laurent et al. 2013, Hattermann et al. 2014, Stewart and Thompson 2015; ). E3SM-HR is with 8 km far from eddy-resolving at this part of the ocean. Please incorporate a sentence or two on the fact E3SM-HR might be 'high-resolution' in terms of Earth System models, but not in terms of resolving the mesoscale in the high latitudes.

Literature mentioned above

Solodoch, A., Stewart, A. L., Hogg, A. M., Morrison, A. K., Kiss, A. E., Thompson, A. F., et al. (2022). How does Antarctic Bottom Water cross the Southern Ocean? Geophysical Research Letters, 49, e2021GL097211. https://doi. org/10.1029/2021GL097211

Nøst, O. A., M. Biuw, V. Tverberg, C. Lydersen, T. Hattermann, Q. Zhou, L. H. Smedsrud, and K. M. Kovacs, 2011: Eddy overturning of the Antarctic Slope Front controls glacial melting in the eastern Weddell Sea. J. Geophys. Res. Oceans, 116, C11014, https://doi.org/10.1029/2011JC006965.

Dinniman, M. S., J. M. Klinck, and E. E. Hofmann, 2012: Sensitivity of Circumpolar Deep Water transport and ice shelf basal melt along the West Antarctic Peninsula to changes in the winds. J. Climate, 25,4799–4816, https://doi.org/10.1175/JCLI- D-11-00307.1.

St-Laurent, P., J. M. Klinck, and M. S. Dinniman, 2013: On the role of coastal troughs in the circulation of warm Circumpo- lar Deep Water on Antarctic shelves. J. Phys. Oceanogr., 43, 51–64, https://doi.org/10.1175/JPO-D-11-0237.1.

Hattermann, T., L. H. Smedsrud, O. A. Nøst, J. M. Lilly, and B. K. Galton- Fenzi, 2014: Eddy-resolving simulations of the Fimbul Ice Shelf cavity circulation: Basal melting and exchange with open ocean. Ocean Modell., 82,28–44, https://doi.org/10.1016/ j.ocemod.2014.07.004.

Stewart, A. L., and A. F. Thompson (2015), Eddy-mediated transport of warm Circumpolar Deep Water across the Antarctic Shelf Break, Geophys. Res. Lett., 42, 432–440, doi:10.1002/2014GL062281.

---

## Author Response (AR2)

**Response to Reviewer 1 for "Southern Ocean polynyas and dense water formation in a high-resolution, coupled Earth System Model" by Jeong et al.**

We thank the reviewer for the additional comments. Please find our responses below (the reviewer's comments are noted in italics and our reference to manuscript line numbers refers to the revised version of the manuscript).

*I thank the authors for thoroughly addressing each comment from my previous review. I only have very few minor comments aimed to further improve the manuscript. I recommend accepting the manuscript for publication after the minor comments have been addressed.*

*Minor comments:*

*L51-57: Solodoch et al. 2022 analyze simulated AABW formation in a global ocean-sea ice model. The study should be mentioned in the introduction.*

We thank the reviewer for introducing this reference. We have added it (line 65).

*L223: Please elaborate on the comparison between the model and Pellichero et al. 2018: Do the results 'compare well' in magnitude, pattern or both?*

We thank the reviewer for this suggestion. We have clarified this on line 220.

*L279: Please use ASF acronym.*

Done.

*Section 2.1 and discussion (L325): Regarding the horizontal resolution of E3SM-HR and resolving the mesoscale. Multiple modelling studies have shown that a resolution of 1-2 km is required to adequately simulate mesoscale eddy activity over the Antarctic continental shelf and slope (e.g., Nost et al. 2011, Dinniman et al. 2012, St-Laurent et al. 2013, Hattermann et al. 2014, Stewart and Thompson 2015). E3SM-HR is with 8 km far from eddy-resolving at this part of the ocean. Please incorporate a sentence or two on the fact E3SM-HR might be 'high-resolution' in terms of Earth System models, but not in terms of resolving the mesoscale in the high latitudes.*

Thank you for noting this. We have first of all corrected a typo in Section 2.1 (line 99: the MPAS resolution varies from 18 km at the equator to 6 km at the poles). We have also added a statement about needing much higher resolution on the continental shelf in order to fully resolve mesoscale eddies on lines 306-309.

**Response to Reviewer 2 for "Southern Ocean polynyas and dense water formation in a high-resolution, coupled Earth System Model" by Jeong et al.**

We thank the reviewer for their helpful and constructive comments. Please find our responses below (the reviewer's comments are noted in italics and our reference to manuscript line numbers refers to the revised version of the manuscript).

*This manuscript is a re-submitted second version of the work. The authors explore dense water formation in coastal and open ocean polynyas around Antarctica using both a low-resolution and a high-resolution version of the fully coupled Energy Exascale Earth System Model (E3SM) forced with 1950 conditions. The work is an important contribution to the modeling society, providing insights into improvements and remaining challenges regarding dense water formation in GCMs. The manuscript is well-written and includes high-quality figures to support the main results. The methodology is sound and builds on state-of-the-art coupled Earth System Model (ESM) development. The manuscript improved much from the first version but will benefit from further revision. Below is a summary of major and minor comments.*

*Main comments:*

*1. The paper will benefit from rewording/restructuring to tell the story more compellingly. For example, a long discussion of what the model cannot do is not appealing to the reader. In the present version, many paragraphs start with what the model cannot do. Then, key findings are given at the end, drowning in all the information about what does not work. Throughout the paper, it would be much more interesting for the reader to focus on what works well, or at least what*

*works better than previous modeling attempts. Then you could briefly overview the remaining challenges and causes/solutions to these challenges.*

We thank the reviewer for this suggestion. We agree that the narrative would benefit from a streamlining of the results presentation. We have followed the specific suggestions mentioned in the minor comments below and also made other minor revisions to the text wherever necessary. Please see, for example, text in blue on lines 148-163, a more streamlined version of the first part of section 3.2, as well as a new version of section 4.2.

*2. The introduction (from line 48) needs restructuring. It would help the reader to give an overview of the current status concerning resolution and the current versions of GCMs. What is currently problematic due to resolution? -And then give more information about why you hope to improve with the HR version of the E3SM-HR. The added information in the discussion from line 325 would be good to give here to set better premises for what you hope to achieve in this paper. The information is partly there, but it could have been more clear.*

We thank the reviewer for this suggestion. Indeed, some reorganization was necessary to pull the relevant information together. We have moved the referenced text out of the Discussion and into the new version of the Introduction. We have also followed one of the reviewer's minor suggestions below to move a paragraph out of Results and into the introduction. Finally, we have pulled together the information about the difficulties that GCM's have in reproducing Southern Ocean processes. Please see the new version of the Introduction section (especially the text in blue), lines 37-77.

*3. The result section on why the winds are too strong and the implications of this is unnecessarily long. You convince me early on, and then you keep arguing for this. You could easily cut parts of this section to keep the reader interested.*

We have reduced section 4.2 a bit to make it more concise (see blue text on lines 247-256). Another reviewer felt strongly about providing more quantitative evidence for the implications of strong winds on the ASC and the cross-shelf stratification, and for this reason, we included the analysis that is summarized in Fig. 8.

***Minor comments:***

*Line 23: Points i and ii combined are key dense shelf water formation mechanisms. You also mention the processes that lead to AABW but only include the more direct process. For example, DSW could also interact with ice shelves or water masses within the ice shelf cavities as part of the AABW formation process.*

We have now added the word 'direct' to line 23. Other indirect mechanisms that impact DSW formation are mentioned in the Discussion section.

*Line 39: What resolution is required to resolve the DSW export along the continental slope?*

The first baroclinic Rossby Radius is less than 20 km over the whole Southern Ocean (e.g., Chelton et al. 1998), with values going down to less than 10 km over parts of the Antarctic continental shelf. Hallberg (2013) suggests resolutions of at least 1-2 km on the shelf in order to be able to represent mesoscale eddies and cross-shore transport properly. The 6 km resolution in E3SM-HR polar regions is adequate, although it is still too low for very coastal regions. We have now added a statement discussing this on lines 306-309.

*Line 145: typo fluxes→flux*

Corrected.

*Line 147: Get to the interesting point right away: "There is more sea ice..."*

Thank you for this suggestion. Please see new sentence in blue on lines 148-150.

*Line 153: Move the general statement "E3SM-HR generally does well..." to the start of the paragraph to help your story flow.*

We have now revised section 3.1 to address this, the following two comments, and the general suggestion in main comment 1. Please see text in blue on lines 148-163.

*Line 157: Get straight to the point "The mean coastal..."*

Please see previous comment and text in blue on lines 148-163.

*Line 159: See major comment 1: The story will improve if you focus on what you do well and why and then compare other features. Start saying that the HR version does much better than the LR version, and tell us why some polynyas are closer than others (MBP, SP, VPB, MP, TNBP, BeP all do well in either/or volume and area)*

Please see text in blue on lines 148-163.

*Line 218: How does this differ if you show the mean over 5 OOP years vs 5 normal years? Since the variation in figure 4 is so large, it would be helpful with some considerations of this.*

We thank the reviewer for this question. To increase the robustness of our WMT results, we opted for considering the regional Weddell Sea WMT as a representation of OOPs effects onto the total

WMT, rather than slicing the model data in time. Please compare the pink and black lines in Fig. 5b as well as panels c and d of the same figure.

*Line 233: Can you comment on why the positive transformation rates occur at lighter densities over the shelf? Is the shelf water generally too light compared with observations, and if so, why?*

We discuss this at the end of Section 4.1 (lines 242-245) as motivation for Section 4.2.

*Line 239: Rewrite the text in the parenthesis.*

Changed to "mostly because less cold" (see line 235).

*Line 250: Please rewrite the first statement in the paragraph to improve readability.*

We have now revised the beginning of section 4.2 (blue text on lines 247-254) substantially, also to follow the suggestions in main comment 3.

*Line 260-272. This is not results but information that belongs to the introduction.*

As mentioned in the reply to main comment 2 above, we have moved this information to the Introduction (see lines 37-54).

*Line 277: Could you start with some general comments before you go into each shelf type? In all shelf types, you find down-sloping isopycnals close to the continental slope. The fresh and dense shelf areas are similar, indicating little/no(?) dense water formation. On the warm shelf, you have upward-tilting isopycnals offshore, but the tilt does not extend onto the shelf.*

Please see text in blue on lines 259-260.

*Line 332: How will land-fast ice alter the dense water formation? Would you get more/less dense water?*

Landfast ice could enhance dense water formation due to the possibly increased divergent motion and associated larger coastal polynyas as mentioned on lines 165-168.

*Line 336: You say you want to include land-fast ice in the future, but how will you fix the too-strong wind problem?*

Thank for this suggestion. We have now added a sentence about this on lines 312-315.

*Figure 1c is hard to read. Could it be an idea to lower the longitudinal resolution of the production data to make it easier to compare the two data sets? If you, for instance, make longitude bins of 10 degrees, the figure would be less crowded. I guess the exact position of the individual polynyas varies from model to observation anyway, and a lower resolution would not be a problem.*

We thank the reviewer for this comment. We would prefer to leave Fig. 1c unchanged.

*Figure 2 would benefit from rescaling the axes on individual polynyas. I generally appreciate similar axes for inter-comparison, but given that RISP is so different, it would be beneficial to zoom in on the others and comment on the different aces for RISP. Panels n and o in figure 2 do not contribute much to the story. I guess RISP is the main contributor to the discrepancies(?) I think you could either give the total numbers or consider plotting stacked bar charts for volume/area for the individual polynyas to display the effect of each polynya on the sum.*

We agree with the reviewer's suggestion and have changed the limits of the y-axis for panels 2a-i and 2k-m to be 0-18, while leaving them unchanged in 2j (we have noted the different scale in the

figure caption). As a result, we think Fig. 2 is now more legible.

*Figure 3 caption: The last sentence seems to be the opposite of what is correct. SIC lower than 15% is masked out.*

Corrected. Thank you for catching this.

*Figure 3g-i: I do not think showing the sensible heat flux in polynya months is necessary. Rather comment on the increase in the polynya area, or say something about the overall contribution to heat loss.*

Thank you for this suggestion. As the reviewer suggested, we removed Fig. 3g-i and commented on the increase in the polynya area on lines 181-182.

*Figure 4: Is it possible to add numbers for dense water production for the various years?*

We believe that the number of simulated years is insufficient to compute the interannual variability of dense water production. For this reason, we decided to focus on the climatological results for this study.

*Figure 8: There is no dense water in either the strong or weak wind composites. I do not think it is necessary to show this figure, but you could add this to the supplementary instead.*

This figure and the description of the analysis behind it (lines 271-279) were added as a response to another reviewer comment that we should provide more evidence that the strong winds in E3SM-HR are impacting the ASC and cross-shelf stratification. We therefore prefer to leave this figure in the main body of the manuscript.